# Chemical trends of deep levels in van der Waals semiconductors

Penghong Ci [1,2], Xuezeng Tian [3], Jun Kang[4], Anthony Salazar[1], Kazutaka Eriguchi[1], Sorren Warkander [1], Kechao Tang[1], Jiaman Liu[1], Yabin Chen[1,5], Sefaattin Tongay[6], Wladek Walukiewicz[2], Jianwei Miao [3], Oscar Dubon[1,2] & Junqiao Wu [1,2]✉

Properties of semiconductors are largely defined by crystal imperfections including native defects. Van der Waals (vdW) semiconductors, a newly emerged class of materials, are no exception: defects exist even in the purest materials and strongly affect their electrical, optical, magnetic, catalytic and sensing properties. However, unlike conventional semiconductors where energy levels of defects are well documented, they are experimentally unknown in even the best studied vdW semiconductors, impeding the understanding and utilization of these materials. Here, we directly evaluate deep levels and their chemical trends in the bandgap of $MoS_2$, $WS_2$ and their alloys by transient spectroscopic study. One of the deep levels is found to follow the conduction band minimum of each host, attributed to the native sulfur vacancy. A switchable, DX center - like deep level has also been identified, whose energy lines up instead on a fixed level across different hosts, explaining a persistent photoconductivity above 400 K.

[1] Department of Materials Science and Engineering, University of California, Berkeley, CA 94720, USA. [2] Materials Sciences Division, Lawrence Berkeley National Laboratory, Berkeley, CA 94720, USA. [3] Department of Physics & Astronomy and California NanoSystems Institute, University of California, Los Angeles, CA, USA. [4] Beijing Computational Science Research Center, Beijing, China. [5] School of Aerospace Engineering, Beijing Institute of Technology, Beijing, China. [6] School for Engineering of Matter, Transport, and Energy, Arizona State University, Tempe, AZ 85287, USA. ✉email: wuj@berkeley.edu

Defects with energies falling within the bandgap may act as a trap or emitter of free charge carriers[1], a site for exciton recombination[2], and a center to scatter electrons or phonons[3]. In conventional semiconductors, native defects such as vacancies introduce levels close to the middle of the bandgap when the material is more covalently bonded, or close to the band edges when the material is more ionically bonded, resulting in the former materials being defect sensitive while the latter materials are relatively defect tolerant[4]. Comparing positions of defect levels across different host materials helps to reveal chemical trends that inform defect models with broad impact. For example, the deep level associated with a given impurity[5] or native defect[6] tends to lie universally at a fixed energy position with respect to the vacuum level even when doped in different semiconductors, which can be used to determine band alignments of the host materials; equilibrium native defects tend to drive the Fermi level toward a stabilization position, and this position with respect to the bandgap can be used as a descriptor of doping propensity and doping limit of the semiconductor[7]; the DX center, an metastable defect switchable between deep and shallow states, dominates the free electron density in III-V semiconductor alloys[8]. It is critical to ask whether such insights and knowledge attained in studying conventional semiconductors are applicable in vdW materials. New effects of defects may emerge because the layered nature of vdW materials allows stronger lattice relaxation as well as new types of defects such as intercalated atoms.

Scanning tunneling microscopy (STM) is able to experimentally visualize various types of defects on the surface and relate these imperfections to electronic structures in vdW crystals[9], in particular for the most abundant native point defects that play a critical role in their electrical[10–13], optical[2], magnetic[14], catalytic[15] and sensing properties[16]. However, STM studies have led to inconsistency on the defect types with transmission electron microscopy investigations, as well as discrepancy in signatures of defect-induced mid-gap states from theoretical calculations[1,3,17–21], largely because of unclear differentiation of STM contrast between the metal and chalcogen sublattices and the complicated convolution of electronic and geometric structures[9]. Furthermore, it shows very limited capability in detecting defects beneath the surface.

In this work, we use deep level transient spectroscopy (DLTS), a high-frequency capacitance transient thermal scanning method[22,23], to characterize electronic structures of the deep traps inside the bandgap of vdW semiconductors, particularly MoS$_2$, WS$_2$ and their alloys, including their energy positions and capture cross sections. Combined with atomic-resolution scanning transmission electron microscopy and first-principles calculations, one of the deep levels determined by DLTS is identified as sulfur vacancies, whose energy position follows the conduction band edge in the host materials, distinct from vacancy defects in traditional group III-V semiconductors. A metastable DX center is identified in these vdW semiconductors, featuring a persistent photoconductivity above 400 K and explaining the chemical trend of native electron concentration in the hosts.

## Results

### DLTS devices and DLTS spectra.
Mechanically exfoliated, multilayer (~ 50 nm) flakes of freshly grown Mo$_{1-x}$W$_x$S$_2$ ($x = 0$, 0.4, 0.7, 1) crystals were made into two-terminal Schottky-Ohmic devices (Fig. 1a). The Schottky contact was formed by dry-stamping freshly exfoliated flakes onto pre-deposited Pt electrodes and confirmed by the $I–V$ and $C–V$ curves shown in Fig. 1e, f, both of which show the n-type conductivity of MoS$_2$. This maximally protects the depletion region at the Schottky contact against contamination and damage[24], as it is at this region where

the deep levels trap and emit charge carriers during the DLTS measurement. The measured total capacitance (Supplementary Fig. 10) is composed of that of the DLTS device (C$_{device}$) and the stray capacitance (C$_{stray}$) connected in parallel. The latter, although with a large value, is insensitive to the external differential voltage (Supplementary Fig. 10), hence the variation of capacitance under the biased voltage indeed probes the former (Fig. 1f).

The depletion width at the Schottky junction (~ 20 nm, the shadow in Fig. 1d), hence the capacitance (Fig. 1c), is initially held constant by a steady-state reverse bias ($V_R = -0.5$ V, stage ①)[23]. An opposite voltage pulse ($V_P$) is then added onto $V_R$, reducing the depletion width (as evidenced by the increased capacitance at less-negative voltage, Fig. 1f), and allowing the traps in the initial depletion region to be filled with free electrons (stage ②)[23]. When the initial, constant bias is restored, the return of the capacitance to the steady-state value is characterized by a transient (stage ③) related to the emission of majority carriers from the deep traps in the material. The capacitance difference within a rate window (between the pre-set t$_2$ and t$_1$ in Fig. 1c)[22] reaches the maximum at a specific temperature. The emission rate ($e_n$) in stage ③ depends exponentially on temperature via the trapping energy level ($E_i$) measured from the conduction band minimum (CBM, $E_{CB}$)[22],

$$\frac{e_n}{T^2} = K\sigma_n \exp\left(-\frac{|E_{CB} - E_i|}{k_B T}\right) \quad (1)$$

where $\sigma_n$ is the capture cross section, and $K$ is a known constant. Arrhenius plots of Eq. (1) at various rate windows (0.5 ms to 20 ms in Fig. 2a) allow extraction of the activation energy of deep levels, $E_{CB}–E_i$. For MoS$_2$ we found two, $0.27 \pm 0.03$ eV (peak A) and $0.40 \pm 0.02$ eV (peak B), as shown in Fig. 2b. The positively valued DLTS peaks (Fig. 2a) indicates that these are majority carriers traps in MoS$_2$[22]. We also measured current transient spectroscopy (CTS, see Supplementary Fig. 1) by recording the current rather than capacitance under the pulsed bias[25], yielding an activation energy of $E_{CB}–E_i = 0.25 \pm 0.02$ eV for MoS$_2$ (Supplementary Fig. 1a), consistent with the peak A in DLTS. We note that for each of the trap energies obtained in this work, at least two devices were measured and all show consistently very similar energy. Thermodynamically, the slope of Eq. (1) corresponds to the change of enthalpy ($\Delta H$), different from the Gibbs free energy $\Delta G$ (= $E_{CB}–E_i$)[26], but the difference can be neglected when electrons are excited from the traps to the conduction band without invoking changes in the bonding configuration (see Supplementary Note 5)[27].

### Determination of sulfur vacancies from STEM and DFT calculations.
To reveal the atomic origin of these deep traps, we have performed first-principles calculations of S single vacancies ($V_S$) in multilayer MoS$_2$, WS$_2$ and their alloys. $V_S$ is chosen because it is the most abundant defect known to naturally occur in these materials[21]. The calculation shows that $V_S$ would introduce a deep-level state with energy of 0.29 eV (for MoS$_2$) and 0.21 eV (for WS$_2$) below the CBM, in good agreement with the DLTS/CTS results. We note that the value of 0.29 eV is also consistent with the calculated $V_S$ energy in MoS$_2$ previously reported in literature[1,3]. Our calculations also confirm that $V_S$ is a deep acceptor, labeled as (0/−)[1,23], not responsible for the natively n-type conductivity of MoS$_2$. The neutral ground state implies its extremely weak Coulomb attraction to electrons, and hence very small capture cross section. $V_S$ defects are directly observed in these materials by scanning transmission electron microscopy (STEM, Fig. 1b and Supplementary Fig. 5)[11], where the density of $V_S$ is directly determined to be $1\sim3 \times 10^{20}$ cm$^{-3}$

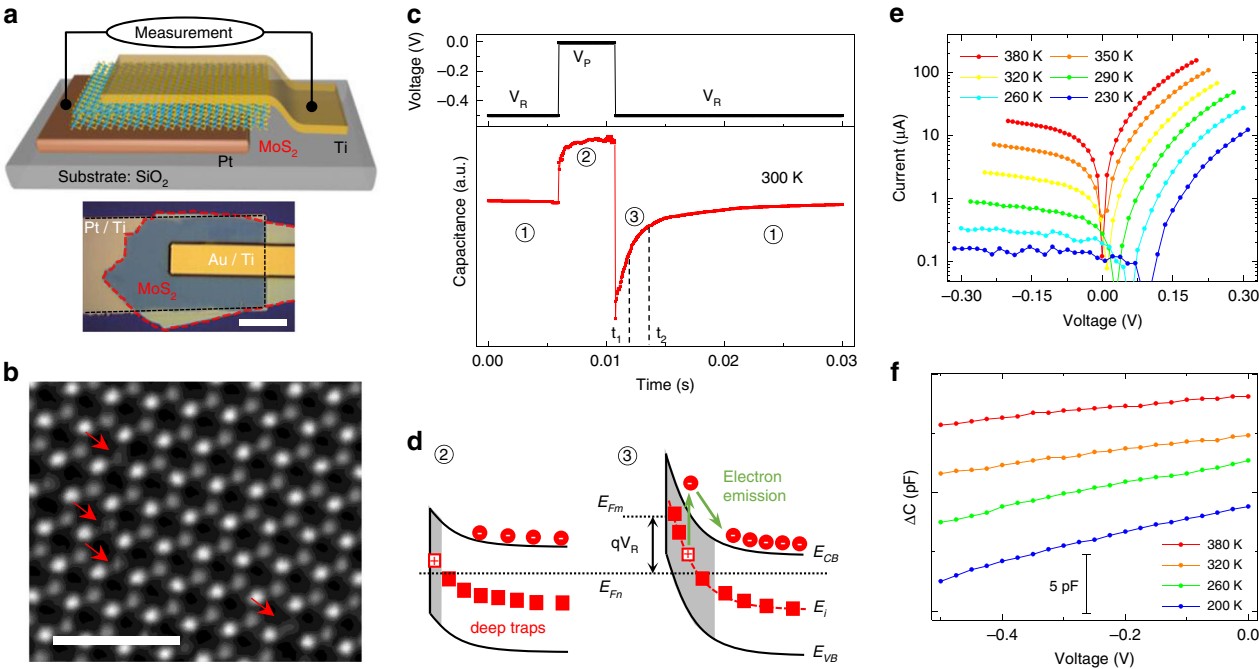

**Fig. 1 Materials and devices for transient spectroscopic study of defects. a** Schematic and optical image (scale bar: 20 μm) of an asymmetric $MoS_2$ device for DLTS, with Schottky contact ($MoS_2$/Pt/Ti) on the bottom and Ohmic contact (Au/Ti/$MoS_2$) on the top. **b** Aberration-corrected STEM image of a monolayer $MoS_2$ exfoliated from the materials used for devices. Red arrows highlight S vacancies ($V_S$). Scale bar, 1 nm. **c** Capacitance transient (bottom) in response to a pulsed change in bias voltage (top). **d** Band bending of the Schottky junction ($MoS_2$/Pt), illustrating the electron trapping (②) and emission process (③) of deep traps in the depletion region (shaded). $V_R$ tunes the Fermi level of the n-type $MoS_2$ ($E_{Fn}$) with respect to that of the metal contact ($E_{Fm}$). **e** & **f**, Temperature-dependent I–V and C–V curves confirming the Schottky-Ohmic contacts.

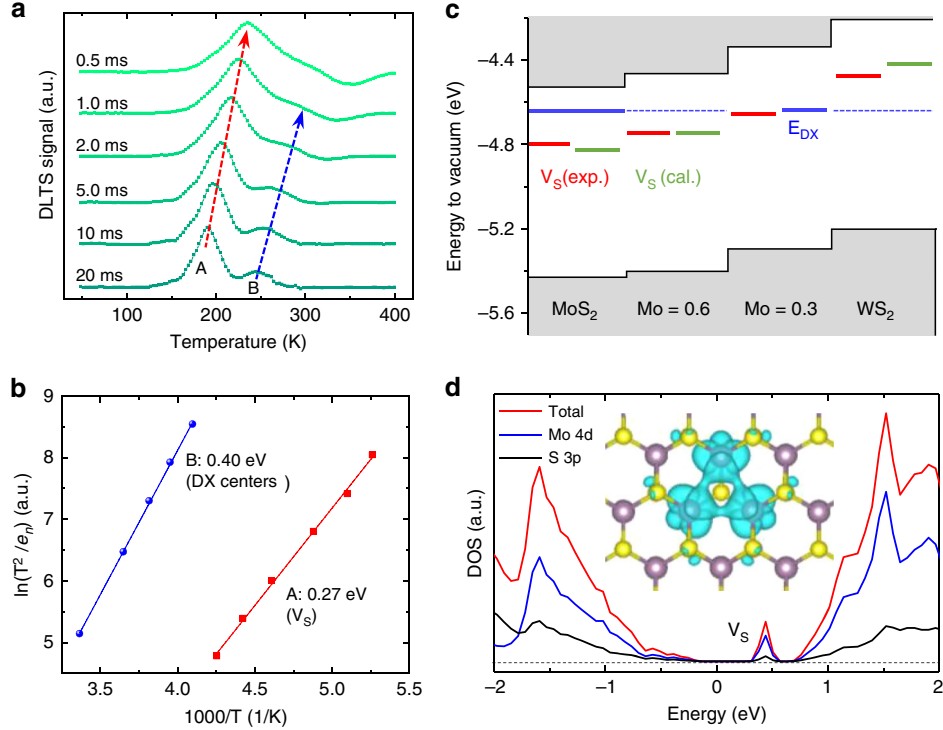

**Fig. 2 Deep levels and their alignment in vdW crystals. a** DLTS signal of a $MoS_2$ device at different rate windows and **b** the resultant Arrhenius plots to determine the activation energies. **c** Conduction and valence band edge alignment calculated with respect to the vacuum level, and positions of deep level experimentally identified in this work. Solid black lines: CBM and VBM in bulk crystals from our DFT calculations; red bars: deep levels attributed to $V_S$ measured by DLTS and CTS; blue bars: DX center levels determined by DLTS and PPC (dashed blue line is guide to the eye); green bars: DFT-calculated $V_S$ levels. **d** density of states (DOS) for multilayer $MoS_2$ with $V_S$. Inset: real space distribution of the wavefunction of $V_S$ state. The purple and yellow spheres represent Mo and S atoms, respectively.

(Supplementary Fig. 4), on the same order of those reported in literature[3,21]. The STEM study also confirms that $V_S$ is the dominant point defects, and no other defects or impurities were detected in the materials. We note that akin to conventional semiconductors, not all of these $V_S$ are electronically active (Supplementary Fig. 6); in fact, deep traps can be highly passivated or compensated, as observed in GaN and GaAs[28,29].

To reveal the chemical trend of the $V_S$ level in different vdW semiconductors, $Mo_{0.6}W_{0.4}S_2$, $Mo_{0.3}W_{0.7}S_2$, and $WS_2$ were also synthesized and then assembled into Schottky devices for similar DLTS/CTS measurements (Supplementary Fig. 1). All of these materials exhibit at least one deep level, akin to the feature A observed in $MoS_2$, with an energy level below the CBM of the host material of $0.29 \pm 0.02$ eV, $0.31 \pm 0.02$ eV and $0.26 \pm 0.04$ eV (red bars in Fig. 2c), respectively. These energy levels are all in good agreement with the DFT calculated $V_S$ levels, as shown by the green bars in Fig. 2c and the refined band structure with $V_S$ in Supplementary Fig. 8.

Some deep levels in different isovalent materials line up at a fixed position with respect to the vacuum level, such as oxygen dopant or Ga dangling bond in different $GaAs_{1-x}P_x$ alloys[6,30]. In contrast, the red bars in Fig. 2c show that as the W fraction increases in $Mo_{1-x}W_xS_2$, the energy level of $V_S$ shifts monotonically toward the vacuum level; that is, the $V_S$ level largely follows the CBM of the host. This is understandable because, as shown in the partial density of states plot in Fig. 2d, the $V_S$ state originates mostly from the 4d (5d) orbitals of the Mo (W) atoms, rather than the S atoms, sharing the same orbital composition as the CBM[31,32]. Following this finding, anion impurities (such as oxygen) substituting S are predicted to create deep levels also about 0.3 eV below the CBM of the host (see Supplementary Fig. 7), because it is known that highly electronegative, substitutional dopants tend to have similar wavefunctions as those of ideal vacancies[6]. The electron capture cross section ($\sigma_n$) of $V_S$ is evaluated from Eq. (1) to be $\sim 3.6 \times 10^{-18}$ cm² in $MoS_2$, using the thermal velocity effective mass (0.57 $m_o$) and effective density of states mass (0.50 $m_o$) obtained from our DFT calculation and literature (see Supplementary Note 1). This value is small but comparable to that of Zn acceptor level in Si and Cu acceptor level in Ge[23,33].

**Persistent photoconductivity and DX center model.** To explore the origin of peak B (0.40 eV) in $MoS_2$ shown in Fig. 2a, we obtained complementary information about deep levels from photoconductivity measurements. Photoconductivity, especially when it is persistent (persistent photoconductivity, PPC), has been used to gauge conduction by charge carriers photo-liberated from certain deep traps[34,35]. Figure 3a shows temperature-dependent dark conductance of a $MoS_2$ flake (~50 nm thick) measured in four-probe geometry (Inset of Fig. 3b). The sample was cooled in darkness from 400 K to 200 K (black data points). It was then exposed to white light for 10 min (blue data point) at 200 K, during which the conductance became two orders of magnitude higher than in the dark. When the light was switched off (dark again) at this low temperature, the conductance dropped slightly, but still stayed >50 times higher than the pristine dark state. The PPC stayed at this level for at least 11 h at 200 K (Fig. 3b). When the sample was warmed up, the conductance stayed at the higher level (red data points) until a temperature of 400 K where it nearly converged to the pristine dark conductance.

Such a PPC effect in response to light exposure and temperature is a direct manifestation of metastability of defect states, and a hallmark of DX centers in semiconductors[34,35]. DX centers, observed in the 1980s in many III-V semiconductors such as AlGaAs, are a special type of localized states resonant

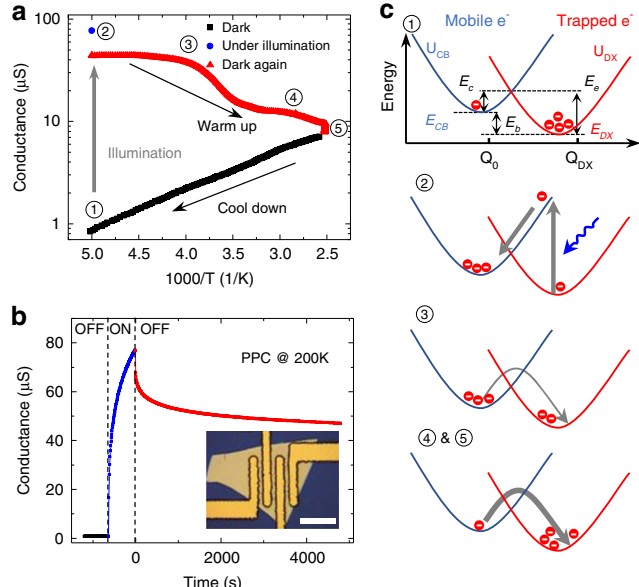

**Fig. 3 Temperature-dependent persistent photoconductivity (PPC) and the DX center model. a** Conductance of a $MoS_2$ device as a function of temperature before, during, and after exposure to light illumination. **b** PPC transient of the device at 200 K. Inset: optical image of a four-probe device for PPC measurement (scale bar: 20 μm). **c** Configurational coordinate diagram (CCD) showing the three energies to characterize the DX center and describe the five PPC processes in **a**.

with the conduction band of the host[8]. In contrast to ordinary deep levels, DX centers are capable of switching into a charge-delocalized, electron-donating state via significant lattice relaxation when triggered by external stimuli, such as light and gate control[8,36,37]. Typically described in the configurational coordinate diagram (CCD) as shown in Fig. 3c, DX centers are characterized by a parabolic coordinate (Q) dependence of DX center energy ($U_{DX}$) intersecting that of the delocalized state ($U_{CB}$)[8]. The displacement along the Q axis between the two minima describes a large lattice relaxation that reflects the metastability of the DX centers. Three energies are thus defined: capture activation energy ($E_c$), which is the energy barrier for the DX center to trap an electron and can be determined from the kinetics of PPC; emission activation energy ($E_e$), the energy barrier to de-trap (emit) an electron, measured via DLTS[8,36]; and energy depth ($E_b = E_{CB} - E_{DX} = E_e - E_c$), which is the ground state energy ($E_{DX}$) measured from the CBM ($E_{CB}$) and can be derived from the temperature dependence of conductance.

As shown in Fig. 3a, c, at the thermal equilibrium state (stage ①), most electrons are trapped in the DX centers. Upon excitation by light with energies above the optical threshold (stage ②)[8,36], electrons in $E_{DX}$ are photo-excited to $E_{CB}$. When the light is off, these electrons stay in $E_{CB}$ and are blocked by the barrier $E_c$ from relaxing back to $E_{DX}$, causing the PPC (stage ③). When temperature rises, more electrons are thermally excited over $E_c$ into $E_{DX}$ (stage ④), eventually recovering to the pristine, dark-state conductivity (stage ⑤). In this study, the PPC effect exists at temperatures up to more than 400 K (upper limit of our equipment). This is in stark contrast to the PPC effect of DX centers discovered in group III-V semiconductors, where it survives only at $T < \sim 140$ K[35,37,38].

The transient PPC curves are plotted in Fig. 4a for a range of temperatures, where non-persistent photocurrent was excluded, dark current was subtracted and the remaining part was normalized by the value at $t = 0$, the moment the illumination

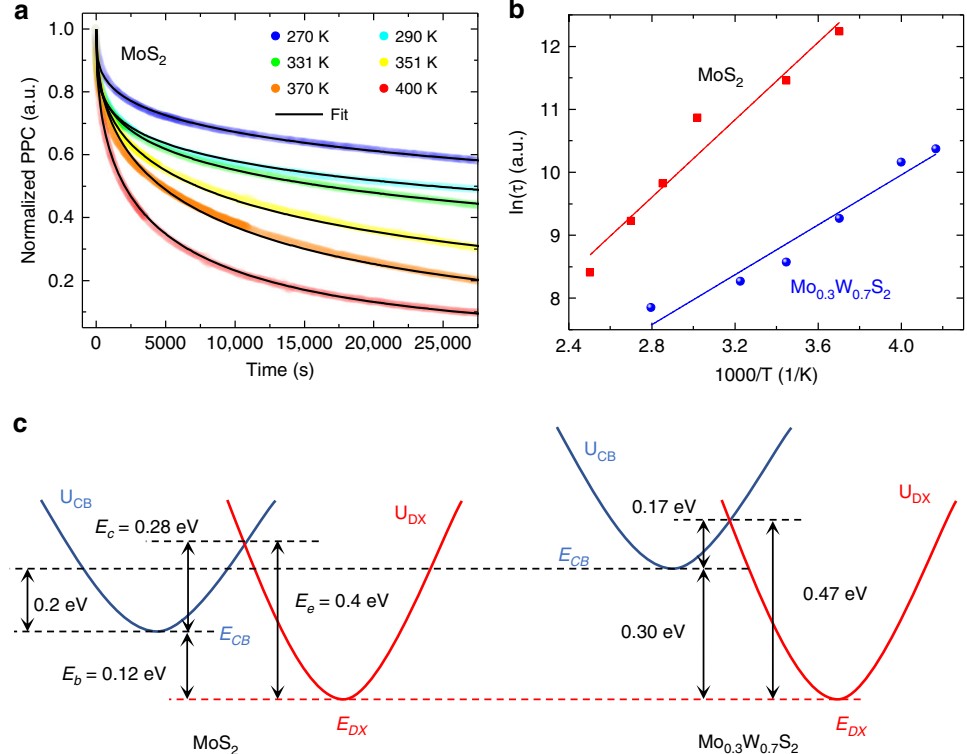

**Fig. 4 DX center levels measured in MoS$_2$ and Mo$_{0.3}$W$_{0.7}$S$_2$. a** Transient normalized-PPC curves at various temperatures for MoS$_2$ and **b** the resultant Arrhenius plots of the time constant for MoS$_2$ and Mo$_{0.3}$W$_{0.7}$S$_2$. Semitransparent points: experimental data; thin solid lines: fitting to Eq. (2). **c** CCD for MoS$_2$ and Mo$_{0.3}$W$_{0.7}$S$_2$, where the energy depth ($E_b$), capture ($E_c$) and emission ($E_e$) energy barriers are determined by temperature dependence of conductance, PPC and DLTS, respectively. The band offset between the two materials is obtained from DFT, resulting in a flat lineup of the DX center energy level ($E_{DX}$) across different host materials. The error range for these energies is estimated to be ± 0.04 eV.

is terminated. Note that in order to reset the initial dark current before taking each of these PPC curves, the samples were kept at 400 K for at least one day in a high vacuum (~$10^{-6}$ torr) to drain the extra electrons in $E_{CB}$. We see that, consistent with the DX center model (Fig. 3), high temperature expedites the kinetics of the PPC decay. Following the treatment in literature, the PPC can be well described by the stretched-exponential equation[34,35]:

$$I_{PPC}(t)/I_{PPC}(0) = \exp[-(t/\tau)^\beta] \quad (2)$$

where $\tau$ is the characteristic decay time constant, $\beta$ is a decay index with a value between 0 and 1. Because of the underlying thermal activation process, the temperature dependence of $\tau$ is related to the trap barrier via $\tau \propto \exp(E_c/k_BT)$[34,35]. Arrhenius plots of the temperature-dependent $\tau$ yield $E_c$ of 0.28 ± 0.02 eV for MoS$_2$ and 0.17 ± 0.02 eV for Mo$_{0.3}$W$_{0.7}$S$_2$ (Fig. 4b and Supplementary Fig. 3). These values are higher than $E_c$ (~0.14 eV) of DX centers reported in the Se-doped AlGaAs system[8], presumably because the layered structure of the vdW materials allows larger lattice relaxation than the tetrahedral structure of AlGaAs. The higher $E_c$ is also responsible for the extension of PPC to much higher temperatures.

The energy $E_b$ ($= E_{CB}-E_{DX}$) characterizes the thermodynamic energy depth of the DX center, and was extracted from Arrhenius plots of the dark conductance of the sample (Supplementary Fig. 3). Values of $E_b = 0.12$ eV and 0.30 eV were found for MoS$_2$ and Mo$_{0.3}$W$_{0.7}$S$_2$, respectively. Adding $E_b$ to $E_c$ gives $E_e$, the emission barrier, of 0.39 eV and 0.47 eV for MoS$_2$ and Mo$_{0.3}$W$_{0.7}$S$_2$, respectively. These values are in very good agreement with the energies of peak B measured in DLTS for MoS$_2$ (0.40 ± 0.02 eV) and Mo$_{0.3}$W$_{0.7}$S$_2$ (0.47 ± 0.02 eV). Therefore, we attribute the peak B measured in DLTS to emission of

electrons from the DX centers. We note that, unlike regular deep levels (such as the V$_S$ state) which have no capture/emission barriers, for DX centers, the Arrhenius plot of the DLTS spectrum extracts the emission barrier $E_e$ (Figs. 2c and 3c), rather than $E_b$ which is the separation of $E_{DX}$ directly measured from the conduction or valence band edges (see more in Supplementary Note 6)[23]. Following the CBM offset of ~0.3 eV between MoS$_2$ and WS$_2$ from our DFT calculation, the CBM ($E_{CB}$) of Mo$_{0.3}$W$_{0.7}$S$_2$ is interpolated to be higher than that of MoS$_2$ by 0.2 eV. Combining all these energy values, the energy of $E_{DX}$ shows an interestingly flat alignment across these two compositions, as plotted in the CCD in Fig. 4c. It is not surprising to see that the $E_{DX}$ position is independent of the material composition because it is also constant for DX centers in AlGaAs across different alloy compositions[8,36,38]: in AlGaAs alloys, $E_{DX}$ is located universally at 3.8 eV below the vacuum level, and does not follow the CBM of the host material (in contrast to shallow defect levels). DX centers act as deep traps that result in different shallow donor doping efficiency in AlGaAs with different compositions[8]; similarly, the chemical trend of energy level of DX centers in the vdW semiconductors can explain the well-known, orders of magnitude higher native free electron density in undoped MoS$_2$ than in WS$_2$, as the DX centers are shallower in the former (details in Supplementary Fig. 9). When they are doped, these deep defects also largely determine the doping efficiency and dopability of these materials, as they can compensate the shallow dopants.

## Discussion

Although our multipronged experiments show clear evidence of DX centers in these vdW semiconductors, elucidation of the

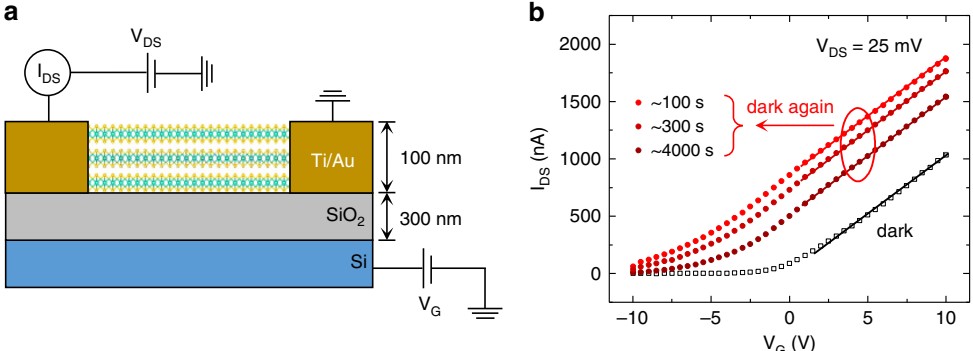

**Fig. 5 Mobility of MoS₂ before and after illumination. a** Schematic of a multilayer MoS₂ field-effect transistor (FET). **b** Transient transfer characteristics for the FET before the illumination (hollow points) and at specific time after the illumination is turned off (solid points). The back gate voltage, $V_G$, is applied to the substrate. The solid lines show the slopes of the $I_{DS}$–$V_G$ curves, corresponding to electron mobility of the channel material in the device.

atomic origin of the DX centers requires further exploration including extensive first-principles calculations. However, the flat alignment of $E_{DX}$ provides a clue. In AlGaAs, the electron wave-function of the DX center is extremely localized on an Al/Ga site surrounded only by and bonded only to the nearest As atoms; therefore, $E_{DX}$ is very insensitive to the change of Al fraction in the alloy[37,39]. Similarly, in $Mo_{1-x}W_xS_2$ alloys where $E_{DX}$ is independent of the cation composition $x$, it is likely that the DX centers neighbor only S atoms, hence are either impurity atoms substituting the cation, or small interstitial atoms bonded to S. For example, a potential candidate would be a defect complex involving hydrogen bonded to S, a dopant inevitably and unintentionally introduced during the growth. Indeed, hydrogen has been proposed to be a possible origin of n-type native conductivity in MoS₂ due to the formation of shallow levels[40].

The decay time constant describes the time it takes for the mobile electrons to be re-trapped by the DX centers, and dictates the relaxation kinetics of the free charge carrier density. The transient conductivity or current in the PPC (Fig. 4a) is assumed to have a similar relaxation kinetics as that of free carrier density, and hence can be used to extract the delay time constant for DX centers. This assumption is typically made in investigation of DX centers in traditional III–V semiconductors as the carrier mobility varies much less than the carrier density and the current is then directly proportional to carrier density[34–36,41,42]. In order to test the validity of this assumption in our case, a multilayer MoS₂ FET was made to determine the evolution of mobility before and after the illumination. Figure 5a depicts a multilayer MoS₂ assembled into the FET which is subjected to a back gate voltage ($V_G$). According to the data in Fig. 5b, we extract the low-field field-effect mobility to be ~16 cm²/(V S) for the MoS₂ channel, based on the expression $\mu = [dI_{DS}/dV_{Gate}] \times [L/WCV_{DS}]$[43], where $W$ and $L$ are the width and length of the channel, respectively, $C$ is the capacitance of the gate dielectric SiO₂ layer, and $V_{DS}$ is the source-drain voltage in the FET. These $I_{DS}$–$V_G$ curves exhibit the same slope before and after light illumination, indicating a constant mobility regardless of the density of free or trapped electrons in the channel, hence validating the extraction of time constant from electrical current via Eq. (2).

It is technically challenging to apply the DLTS to monolayers of vdW semiconductors, owing to expected high leakage current and issues arising from the sub-depletion width thickness. However, the deep levels we quantified for thick layers are expected to be applicable to monolayers and few layers. This is because the very weak interlayer vdW coupling only modulates the conduction and valence band edges, transitioning the material from direct bandgap in monolayers to indirect bandgap in the

bulk, while hardly affecting the entire band structure[32]; on the other hand, the spatially very localized wavefunctions of deep levels do not hybridize with the conduction or valence band edges, which is in contrast to shallow defects whose wavefunctions are composed of entirely the band edge states. For example, regarding monolayer MoS₂, first-principles calculations predicted that the $V_S$ deep level lies about 0.5 eV below the CBM at the $K$ point in the Brillouin Zone[1,3,17,21,44], which is in good agreement with the 0.27 eV below the CBM at the $Q$ point in multilayer MoS₂ quantified in this study, considering the 0.2 eV CBM offset between monolayer and bulk MoS₂[45,46].

Our work determines energy levels and chemical trends of the most abundant native defects in MoS₂, WS₂ and their alloys. These energy levels offer quantitative references for both applications that are limited by defects such as transistors[10,24] and light emitting devices[2], as well as applications that are facilitated by defects such as catalysis[15] and sensors[16]. We also discover metastable and switchable, DX center-like defects in these vdW materials at temperatures above 400 K, in contrast to those in other semiconductors that exist only at $T < 140$ K[8,38]. As a result, practical device applications may be developed from the DX centers in vdW materials, such as nonvolatile memory based on a single defect. These defects may provide a platform for study of electron-phonon coupling, electron correlation, and many-body physics such as negative-U effects in quasi-two-dimensional crystals[30].

## Methods

**Materials preparation**. The vdW bulk crystals were synthesized using the flux zone technique without using transporting agent precursor, in order to reduce contamination[47]. The growth starts with 6N-purity, commercially available 300 mesh amorphous powders of molybdenum and/or tungsten and pieces of sulfur. Further electrolytic purification was necessary to eliminate magnetic impurities commonly found in metal powders, and secondary ion mass spectroscopy (SIMS) was used to test the purity. Powders were mixed at stoichiometric ratios, sealed under $10^{-7}$ torr pressure in quartz ampoules, and annealed up to 800 °C for 10 days. The polycrystalline products were collected and resealed again. In the second formation process, a small temperature differential (~15 °C) was created at high temperatures to thermodynamically drive the reactions. The crystallization process was slow and the entire growth was completed in a three-month time frame.

**Device fabrication**. Multilayer (~50 nm thick) MoS₂, WS₂ and their alloys were mechanically exfoliated from bulk crystals. For DLTS/CTS experiments, these samples were transferred onto Pt/Ti (45/10 nm) bottom electrodes[10], followed by photolithography, and electron beam evaporation of 20-nm Ti and then 80-nm Au as the top electrodes, and lift-off. In this way, the vdW flake is sandwiched by Pt (Schottky) metal at the bottom, and Ti (Ohmic) metal at the top. For PPC measurements, four-probe metal leads (Au (80 nm)/Ti (20 nm), Ti at bottom) were deposited onto exfoliated samples. The devices used SiO₂ (300 nm)/Si as the substrate.

**Electrical measurements**. A deep level transient spectrometer (Sula Technologies) was used to measure DLTS, CTS, CV, and IV curves in Figs. 1 and 2. In this instrument, the emission rate is set as $e_n = 1/(D \times \Delta t)$, where $\Delta t = t_2 - t_1$ is the preset time difference in Figs. 1c and 2a, and $D$ is a constant representing the delay factor, 1.94 and 4.3 for the DLTS and CTS measurements, respectively. In the capacitance test, including CV and DLTS, an A.C. voltage with an amplitude of ~60 mV and frequency of 1 MHz was superimposed onto the D.C. reverse bias. For the PPC measurements, four-terminal transport characteristics were measured by applying a DC bias to the outer channel and recording the current using a current amplifier and the voltage drop across the inner channel using a voltage amplifier. Optical illumination for the PPC was by a convection-cooled 30-Watt illuminator (Fiber-Lite 190).

**STEM characterization**. Mechanically-exfoliated monolayer $MoS_2$ was transferred from $SiO_2$ surface to TEM grids (Quantifoil R2/2) by selective etching of the $SiO_2$ in 49% hydrofluoric acid. Images were acquired from different regions of the monolayer $MoS_2$ using a Nion UltraSTEM 100 aberration-corrected STEM in ADF-STEM mode with $E = 70$ kV. The beam convergence semi-angle was 30 mrad and the detector collection angle was in the range of 30-300 mrad, where a small detector inner angle was chosen to reduce the electron dose. The energy spread of the electron beam was 0.3 eV. To reduce the total electron dose, images were measured with a beam current of 15 pA and a dwell time of 84 μs per image, which correspond to a total electron dose $4.7 \times 10^5$ e⁻/Å². The ADF-STEM images contain a mixture of Poisson and Gaussian noise and were denoised by the block-matching and 3D filtering (BM3D) algorithm[48], from which S vacancies were identified. It has been reported previously that a 80 keV electron beam induces S vacancies in $MoS_2$ with a rate of $3.45 \times 10^8 - 3.36 \times 10^9$ electrons per S vacancy[3,21]. As we used a 70 keV electron beam, the vacancy formation rate in our experiment should be $>3.45 \times 10^8$ electrons per S vacancy. From the total electron dose used in our experiment, we estimated the electron beam induced S vacancy density in our sample was $<2 \times 10^{20}$ cm⁻³. Since we observed a S vacancy density of $3 \times 10^{20}$ cm⁻³ in the $MoS_2$ sample, we concluded that the native S vacancy density was $>1 \times 10^{20}$ cm⁻³, which is in agreement with that of exfoliated undoped $MoS_2$ samples[21].

**DFT calculations**. The calculations were performed using the Vienna ab initio simulation package (VASP) with the projector-augmented wave method[49,50]. The generalized gradient approximation of Perdew-Burke-Ernzerhof (GGA-PBE) was adopted for the exchange-correlation functionals[51]. The energy cutoff for the plane-wave expansion was set to 350 eV. Structure relaxation was stopped when the force on each atom was smaller than 0.01 eV/Å. The van der Waals interaction was included by using the correction scheme of Grimme[52].

For defect calculations in bulk MX2, we employed $5 \times 5 \times 1$ supercell, where a tilted c lattice vector was adopted, with $c = c_0 + 2a_0 + 2b_0$, where $a_0$, $b_0$, and $c_0$ are the primitive cell lattice vectors. As discussed in previous studies[1], this improves the convergence of total energies with respect to cell size. The k-point sampling is $2 \times 2 \times 2$. The defect charge-transition energy level $\varepsilon(q/q')$ corresponds to the Fermi energy $E_F$ at which the formation energy for a defect α with different charge state q and q′ equals with each other. It can be calculated by[53]:

$$\epsilon(q/q') = [E(\alpha, q) - E(\alpha, q') + (q - q')(E_{VB} + \Delta V)]/(q' - q).$$

Here E(α,q) is the total energy of the supercell containing the defect, and $E_{VB}$ is the valence band maximum (VBM) energy of the host material. The potential alignment correction term $\Delta V$ is added to align the VBM energy in systems with different charged states. It is calculated by the energy shift of the 1s core-level energy of a specified atom (which is far away from the defect site) between the neutral defect and charged cases. For $Mo_{1-x}W_xS_2$ alloys, different S vacancy sites have different local environments. The number of surrounding Mo and W atoms varies, resulting in four types of $V_S$. We calculated the charge-transition levels for each type, and then carried out an average according to the concentration of different types to obtain the final charge-transition level.

## Data availability
The data that support the plots in this paper are available from the corresponding author upon reasonable request.

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

## Acknowledgements

This work was supported by the Electronic Materials Program funded by the Director, Office of Science, Office of Basic Energy Sciences, Materials Sciences and Engineering Division, of the U.S. Department of Energy under Contract No. DE-AC02-05CH11231. The device fabrication was partly supported by the Center for Energy Efficient Electronics Science (NSF Award No. 0939514). J.M. and X.T. acknowledge the support by the US Department of Energy, Office of Science, Basic Energy Sciences, Division of Materials Sciences and Engineering under award DE-SC0010378 and by an Army Research Office MURI grant on Ab-Initio Solid-State Quantum Materials: Design, Production and Characterization at the Atomic Scale (18057522). We are grateful for Prof. Mary Scott and Dr. Yaqian Zhang for assistance in TEM, and Dr. Muhua Sun for drawing the schematic of the DLTS device.

## Author contributions

P.C., O.D. and J.W. conceived this project. P.C. fabricated DLTS and PPC devices and completed the measurements, with the assistance from A.S., K.E., S.W., K.T., J.L., Y.C. and O.D. X.T. and J.M. contributed to atomic-resolution STEM imaging. S.T. grew the bulk MoS2, WS2 and alloys. J.K. performed DFT calculations. P.C., J.K., W.W. and J.W. analyzed the results. All authors discussed and contributed to the preparation of the manuscript.

## Competing interests

The authors declare no competing interests.
