## [Peer Review File · Nature Communications]

REVIEWER COMMENTS

Reviewer #1 (Remarks to the Author):

This paper investigates deep levels in the band gap of Van der Waals (vdW) semiconductors such as MoS₂, WS₂ and their alloys, associated with defects. By using transient spectroscopy measurements the authors are able to directly evaluate the energy of these levels and attribute them to specific defects. A deep level is associated with sulphur vacancy. A DX center is also identified which is a novel aspect of this work. The original conclusion is that the DX center is persistent at temperatures above 400 K, in contrast to other semiconductors where these states that exist only at temperatures lower than 140 K. This is of interest to others in the community as well as in wider fields because these states can be used for practical applications in optoelectronics, memories and sensing.

The work is convincing, but in my opinion some revision of the manuscript is required to strengthen the way this paper will influence thinking in the field, given the large amount of available papers on 2D materials.

A major concern is that the introduction seems to ignore the existing vast experimental literature on the subject of defects in the bandgap of vdW semiconductors. Defects in the bandgap of MoS₂, WS₂ and other similar materials have been investigated experimentally by others. I appreciate that DLTS is a technique that allows to directly evaluate the energy of these levels, but it would be more beneficial if this was more evident from the introduction. There are other techniques that have been used to study these defects and so the benefits of using DLTS over these other techniques should be included. For example, electron tunneling and optical spectroscopy measurements have been employed in [Nature Communications volume 10, Article number: 3825 (2019)] to investigate mid-gap states from vacancies in similar materials to those presented in this manuscript. It is not clear what are the novelty aspects of this work that justifies publication in Nature Communications. What new information is provided that has not been revealed by other works. Is this paper just confirming what has been already known indirectly from other techniques or is there more contribution to this field by directly evaluating these states by DLTS?

The motivation given for this work "It is critical to ask whether such insights and knowledge attained in studying conventional semiconductors are applicable in vdW materials" does not sound very novel as this question has already been addressed by previous papers. I would rather suggest to emphasize the DX center and the novel features obtained in vdW materials as opposed to other semiconductors.

A minor point regarding the introduction is that more up to date literature including experimental papers should be cited. For example, in [Adv. Mater. 2017, 29, 1605598] the authors studied trapped charges on the defects with energies falling within the bandgap of similar materials and their role in the operation of field-effect transistor based on vdW semiconductors. In ref [Nature Communications volume 10, Article number: 4133 (2019)] it was shown that robust trap effect can be introduced in via the synergistic effect of vacancies and substitutional atoms

I also have some technical questions below.

The paper presents studies on multilayer flakes of ~50nm.

Why such thick layers, it would have been more interesting to study the 2D limit of these materials.

Do these states depend on the thickness of the multilayers?

What happens in the atomically thin limit of monolayer and few layer (e.g. 2,3 layers) materials?

The value of the V_s energy in 50nm MoS₂ is compared to values for 1-3 layers reported in the literature. Is this justified?

In ref [AIP Advances 9, 015230 (2019)] the authors used DLTS to study the defect states in MoS₂, This should be included in the comparison with the energy reported in the literature.

Reviewer #2 (Remarks to the Author):

The authors report a study based on spectroscopy techniques (DLTS/CTS) carried out on vdW materials which are definitely promising materials for future applications. A dominant acceptor state is observed and correlated to the vacancy and a metastable state postulated to be responsible of the persistent photoconductivity is proposed. Referring to previous studies carried on III-V compounds the authors call the metastable defect DX center.

Although the approach is not new, the results constitute an interesting contribution. However, in its current form the present manuscript is not suitable for publication in nature communication. Below are the points of concern.

From line 87 to 94 the authors give some quantitative information about the energy positions of the levels (attributed to V and DX defects) with respect to the conduction band minimum. The values are inconsistent with the positions depicted in Fig. 2c (red and blue bars). DX-level (blue bar) must be deeper than V's (red bar).

In supporting information (SI), the authors give relation S9, linking the depletion width to various parameters. Relation S9 assumes that the Schottky barrier is abrupt, meaning a uniform free carrier distribution. This is not obvious when one looks at Fig. 1f. A plot $1/C^2$ versus $|V_R|$ would have been useful. Its linearity or non-linearity would inform about the distribution of free carriers. In that case why do the authors rely on literature to give a value to free electron concentration N ? It can be extracted directly from C-V data. More than that a study of these data at various temperatures as done in Fig. 1 would be very helpful to understand compensation versus passivation, invoked in the manuscript. Also, the same study but as a function of frequency would inform about the charge states. But then a technical issue is raised. What is the oscillation level (V_{osc}) superimposed to the constant V_R , necessary to carry capacitance measurements? In the present case the depletion region seems so tiny (~ 21 nm) that a very small value of V_{osc} is required to have meaningful data. What is this value? Finally, the depletion is so small that Debye incursion cannot be neglected which should certainly affect the shape of the emission transient from which DLTS data are extracted. A serious deviation from pure exponential is very likely. But not only. Do the authors know the mobility of free carriers. In case of low mobility, multi-trapping process is not impossible which should also affect the shape of the transient. All these aspects would have consequences on data analysis.

It is not clear whether the condition for an easy analysis of DLTS spectra, that is $N_T/N_D \ll 1$, is fulfilled or not. N_T being the electrically active concentration of vacancies or DX centers, and N_D the free electron concentration if one assumes that all intentional or non-intentional dopants are ionized leading to N cited above. The authors should give an estimation of this ratio starting from an estimation of the electronically active concentration of vacancies.

The I-V characteristics displayed in Fig. 1e show that at high temperatures the authors must be cautious. The conditions of meaningful DLTS measurements may break down. One of the major assumptions in DLTS is that during the emission process the only carriers present in the depletion region are those coming from the traps. Here, the authors may have to consider those participating in the leakage current coming from all parts of the device.

Fig. 2 and lines 121 to 142 in the manuscript and Fig. S1 discuss the results collected from 4 samples with different alloy compositions. Fig. S1 is specially confusing as it is not clear which DLTS/CTS spectra corresponds to what alloy composition. On the other hand, if we refer to Fig. 2 and Fig. S1d, the enthalpy of ionization attributed to the vacancy is clearly lower in MoS₂ than in [Mo]_{0.3}W_{0.7}S₂. However, the authors claim that the vacancy level follows the CBM for all studied compositions. Do they mean the vacancy level is pinned to CB? Clarifications are needed.

The deeper level in Fig. S1d reflects clearly the alloy disorder at the atomic level. The peak is

wider and asymmetric in contrast with what DLTS or CTS models predict in a homogeneous material. Conventional DLTS is of course unable to sort out the various structures hidden by the broadening and corresponding to various local alloy chemical disorder. Laplace DLTS could help in some restricted conditions. This leads to claim that the energy extracted from the Arrhenius law is an average and the extracted (not measured) capture cross section is in such cases meaningless.

The part dedicated to PPC results is clearly detailed and seems convincing. However, the link of the emission energy E_e to the barely seen peak B in Fig. 2a is puzzling to say the least. All data resulting from PPC measurement seem to indicate a very significant concentration of the metastable DX center that the authors attribute to PPC signal. At 250K corresponding to the lowest rate window, the peak is so weak that it cannot be linked to PPC, unless the weak signal corresponding to peak B has a physical meaning.

For the clarity of the paper, it would be useful to discuss briefly the structure of the bandgap (direct vs indirect) for both MoS₂ and WS₂. In relation to this, the behavior of the band gap as a function of alloying would also be worth discussing as it has fundamental impact on the position of levels in the band gap discussed in the manuscript.

Regarding Note 1 in supporting information

Although the relation of the emission rate used by the authors (equation (1)) is widely accepted in literature, the activation energy has no physical meaning if not expressed in thermodynamic framework. The appropriate energy is the Gibbs free energy which is split into the enthalpy appearing in the exponential and the entropy which appears in the pre-factor. Two consequences result; (i) the emission enthalpy should be temperature dependent except in the case of a strict pinning to the respective allowed band (here the conduction band). It is however understandable that the narrow temperature range allowed by the accessible rate windows might make such a study difficult. This is why one consider the enthalpy roughly temperature independent. But, unless the capture cross section is measured, its extraction from the pre-factor, which is got from the intercept at $T=\infty$, should include the entropy factor. Therefore, the extrapolation leads to the product $e^{(\Delta S_n/k)} \sigma_n$ and not σ_n , where k stands for Boltzmann constant and ΔS_n for the emission entropy. In the present context the authors overestimate the capture cross section by a non-negligible factor $e^{(\Delta S_n/k)}$. One may expect the vibrational modes to be more sensitive to trapping – emission in vdW materials than in conventional covalent materials. The authors mention in line 53-54 of the manuscript that stronger lattice relaxations are to be expected in vdW materials. They should thus comment on this with respect to ΔS_n . If the authors decide to measure the capture cross section they must be aware of non-linearities due to very narrow depletion region for which Debye incursion of free carriers can no longer be neglected.

Reviewer #3 (Remarks to the Author):

Defects are important for atomically thin 2D materials. It is a topic that is not fully investigated. However, the authors are investigating the bulk layered materials instead of single-layer atomically thin 2D materials. More importantly, the reviewer doesn't believe that the model of DX defects is realist unless the authors fully address the following points:

1. If the model is real, the authors should perform photo Hall measurements to verify whether it is the electrons contributing to the persistent photocurrent.
2. Based on the model, how to explain the long rise time in Fig3.b (blue line)?

3. The DX concentration is unlikely high enough to induce the high photoconductivity. The authors can roughly estimate the concentration of DX defects from the photoconductivity.

4. It is unclear the wavelength of excitation light used in the experiments. The author should illuminate the sample with photon energy lower than the bandgap so that only electrons in the DX defects can be excited to the conduction band.

5. The authors should thoroughly investigate the material itself. It is unclear whether the semiconductor is n-type or p-type and what is the conductivity.

6. Lastly, the authors actually investigate the bulk layered materials. The bandgap and band structure will change as the materials become atomically thin. The information on the defect energy levels found in this manuscript may not be applicable to the defects in atomically thin 2D materials.

Point-by-point response to Reviewer #1's comments:

Overall comment: *This paper investigates deep levels in the band gap of Van der Waals (vdW) semiconductors such as MoS₂, WS₂ and their alloys, associated with defects. By using transient spectroscopy measurements the authors are able to directly evaluate the energy of these levels and attribute them to specific defects. A deep level is associated with sulphur vacancy. A DX center is also identified which is a novel aspect of this work. The original conclusion is that the DX center is persistent at temperatures above 400 K, in contrast to other semiconductors where these states that exist only at temperatures lower than 140 K. This is of interest to others in the community as well as in wider fields because these states can be used for practical applications in optoelectronics, memories and sensing.*

The work is convincing, but in my opinion some revision of the manuscript is required to strengthen the way this paper will influence thinking in the field, given the large amount of available papers on 2D materials.

Response:

We appreciate the reviewer's acknowledgment of the importance of our work.

Q1: *A major concern is that the introduction seems to ignore the existing vast experimental literature on the subject of defects in the bandgap of vdW semiconductors. Defects in the bandgap of MoS₂, WS₂ and other similar materials have been investigated experimentally by others. I appreciate that DLTS is a technique that allows to directly evaluate the energy of these levels, but it would be more beneficial if this was more evident from the introduction. There are other techniques that have been used to study these defects and so the benefits of using DLTS over these other techniques should be included. For example, electron tunneling and optical spectroscopy measurements have been employed in [Nature Communications volume 10, Article number: 3825 (2019)] to investigate mid-gap states from vacancies in similar materials to those presented in this manuscript. It is not clear what are the novelty aspects of this work that justifies publication in Nature Communications. What new information is provided that has not been revealed by other works. Is this paper just confirming what has been already known indirectly from other techniques or is there more contribution to this field by directly evaluating these states by DLTS?*

Response:

This is an excellent suggestion. Indeed, scanning tunneling microscopy (STM) made it possible to experimentally visualize the various types of defects on the surface and correlate these imperfections with their electronic structures in vdW crystals¹. However, STM studies have so far led to inconsistent explanations on the defect types with transmission electron microscopy (TEM) investigations, as well as discrepancy of signatures of defect-induced mid-gap states from theoretical calculations²⁻⁸. Furthermore, it is incapable of detecting defects beneath the surface that is intrinsic to bulk vdW materials and not affected by surface adsorbents and reconstruction. Optical microscopy, on the other hand, does not work for detection of defect sites with non-radiative recombination processes, such as the Shockley–Read–Hall process, a dominant recombination process in semiconductors.

In light of these limitations, we demonstrate that deep level transient spectroscopy (DLTS), a high-frequency capacitance transient thermal scanning method^{9,10}, can provide clear electronic structures of traps inside the vdW semiconductors, including their energy level positions and capture cross sections. More importantly, we studied dominant deep levels in the full composition of MoWS₂ alloys, such that the chemical trends of these native sulfur vacancies were understood for the first time. This predicts behavior of other anion impurities induced deep levels and hence paves the way for optoelectronic applications. In addition to the energy position of deep states, DLTS also extracts their capture cross section, an important physical property for carrier thermal emission, which cannot be measured by STM. Lastly, in this study, metastable DX centers are identified in vdW semiconductors for the first time, explaining the persistent photoconductivity above the surprisingly high 400 K.

We have included this discussion of the detection of deep levels by STM into the introduction (Page 2, Paragraph 2), and cited this reference (new Ref.20) mentioned by the reviewer.

Q2: The motivation given for this work "It is critical to ask whether such insights and knowledge attained in studying conventional semiconductors are applicable in vdW materials" does not sound very novel as this question has already been addressed by previous papers. I would rather suggest to emphasize the DX center and the novel features obtained in vdW materials as opposed to other semiconductors.

Response:

As highlighted in the title, we focus on the chemical trend of deep levels by systematic DLTS measurement of the whole compositions MoWS₂ alloys, which remains elusive so far and unaddressed. Our DLTS results, associated with STEM images and DFT calculation, demonstrate that, in contrast to traditional semiconductors, sulfur-vacancies induced deep level follows the conduction band minimum of each host, which has not been explored by previous studies.

Following the reviewer's suggestion to emphasize on the DX centers, in the revised version, we measured charge mobility of MoS₂ before and after the light illumination to liberating the trapped charges. The constant value of mobility further validates our model in extracting the delay time constant and the resultant carrier barrier during the PPC process.

We have added a figure (new Fig. 5) and related discussion in the revised manuscript (Page 14, Paragraph 1).

Q3: A minor point regarding the introduction is that more up to date literature including experimental papers should be cited. For example, in [Adv. Mater. 2017, 29, 1605598] the authors studied trapped charges on the defects with energies falling within the bandgap of similar materials and their role in the operation of field-effect transistor based on vdW semiconductors. In ref [Nature Communications volume 10, Article number: 4133 (2019)] it was shown that robust trap effect can be introduced in via the synergistic effect of vacancies and substitutional atoms

Response:

These two papers studied the critical role of deep defects in the performance of TMD

devices, while not figuring out the atomic origin and doping type of these deep levels, nor measuring their energy positions and other fundamental physical parameters. In contrast, our study explores these key properties of deep states, vital to the design of optoelectronic devices and precise predictions of materials performances. Nevertheless, we agree that these two previous works should be cited.

Both of the two papers are now cited (new Ref.12 and 13) in the revised version.

Q4: *I also have some technical questions below.*

The paper presents studies on multilayer flakes of ~50nm. Why such thick layers, it would have been more interesting to study the 2D limit of these materials. Do these states depend on the thickness of the multilayers? What happens in the atomically thin limit of monolayer and few layer (e.g. 2,3 layers) materials? The value of the V_s energy in 50nm MoS₂ is compared to values for 1-3 layers reported in the literature. Is this justified?

Response:

Despite the transition of bandgap from indirect in multilayer transition metal dichalcogenides (TMDs) to direct in monolayer ones^{11,12}, physical properties of V_s defects and DX centers discovered in multilayer vdW semiconductors are likely applicable for monolayer and few-layer samples. This is because the very weak interlayer vdW couplings only modulate the band edges, hardly affecting the rest of the TMD band structure¹³, while spatially localized wavefunctions of deep levels do not only hybridize with the conduction or valence edges (which is in contrast to shallow defect states whose wavefunctions consist entirely of the band edge wavefunctions). For example, regarding monolayer MoS₂, first-principles calculations predicted that the V_s deep level lies about 0.5 eV below the CBM at the K point of Brillouin Zone^{2,6-8,14}, which is in good agreement with the 0.27 eV below the CBM at the Q point of multilayer MoS₂ determined in this study, considering the 0.2 eV CBM band edge offset between monolayer and multilayer samples^{15,16}. It is reasonable to extend the properties of deep defects attained from multilayers in this study to monolayer ones.

DLTS detects the capacitance variation within a given time window across the sample thickness driven by bias voltage, so ultrathin TMDs (such as monolayers) would introduce high leakage current and hence result in inability or unreliability of capacitance measurements. Meanwhile, the depletion capacitance of Schottky junction requires the sample to be thicker than the depletion width, ~ 20 nm (Fig. S10) for MoS₂, so multilayer TMDs (> 50 nm) were studied in this manuscript.

We have included this discussion in the revised manuscript (Page 14, Paragraph 2).

Q5: *In ref [AIP Advances 9, 015230 (2019)] the authors used DLTS to study the defect states in MoS₂, This should be included in the comparison with the energy reported in the literature.*

Response:

After reading this reference carefully, it is clear that it merely mentioned that DLTS is an effective method to study the deep levels in its abstract and introduction; The authors did not really carry out DLTS, and instead used hysteretic gate transfer

characteristics of FETs to analyze the deep states indirectly. In contrast, in our case, we completed the DLTS experiments to quantify the in-gap deep levels of TMDs.

Point-by-point response to Reviewer #2's comments:

Overall comment: *The authors report a study based on spectroscopy techniques (DLTS/CTS) carried out on vdW materials which are definitely promising materials for future applications. A dominant acceptor state is observed and correlated to the vacancy and a metastable state postulated to be responsible of the persistent photoconductivity is proposed. Refereeing to previous studies carried on III-V compounds the authors call the metastable defect DX center.*

Although the approach is not new, the results constitute an interesting contribution. However, in its current form the present manuscript is not suitable for publication in nature communication. Below are the points of concern.

Response:

We thank the reviewer for reviewing our manuscript and considering our results interesting. Please find our point-to-point replies to your suggestions as follows.

Q1: *From line 87 to 94 the authors give some quantitative information about the energy positions of the levels (attributed to V and DX defects) with respect to the conduction band minimum. The values are inconsistent with the positions depicted in Fig. 2c (red and blue bars). DX-level (blue bar) must be deeper than V's (red bar).*

Response:

There is a misunderstanding. Unlike regular deep levels which have no capture/emission barriers, such as the V_S state, for DX centers the Arrhenius plot of the DLTS spectrum extracts the emission barrier, which is NOT directly measured from the conduction or valence band edges. This is because, in the case of DX centers, the energy barrier, E_c in the configurational coordinate diagram (CCD, Fig. R1), must be overcome in order for an electron to be trapped by defects, hence leading to a strongly temperature-dependent capture cross section¹⁰,

$$\sigma_{n,DX} = \sigma_{\infty} \exp\left(-\frac{E_c}{k_B T}\right). \quad (R1)$$

Combining Eq. (R1) and (1) gives

$$\frac{e_n}{T^2} = K \sigma_{\infty} \exp\left(-\frac{|E_{CB} - E_i| + E_c}{k_B T}\right), \quad (R2)$$

where $|E_{CB} - E_i| + E_c$ is equal to the emission energy, E_e in the CCD (Fig. R1) without considering the entropy change, and E_i is E_{DX} .

Figure R1: Configurational coordinate diagram (CCD) showing the three energies (E_c , E_b and E_e) to characterize the DX center.

To sum up, the DLTS spectrum measures the activation energy or binding energy (E_b) for normal defects such as the V_s states, while for DX centers, DLTS yields the emission energy (E_e), the summation of binding energy (E_b) and capture barrier (E_c).

This discussion has been added in note 6 of Supplementary information and the revised manuscript (Page 12, Paragraph 1).

Q2: In supporting information (SI), the authors give relation S9, linking the depletion width to various parameters. Relation S9 assumes that the Schottky barrier is abrupt, meaning a uniform free carrier distribution. This is not obvious when one look at Fig. 1f. A plot $1/C^2$ versus $|V_R|$ would have been useful. Its linearity or non-linearity would inform about the distribution of free carriers. In that case why do the authors rely on literature to give a value to free electron concentration N ? It can be extracted directly from C-V data. More than that a study of these data at various temperatures as done in Fig. 1 would be very helpful to understand compensation versus passivation, invoked in the manuscript.

Response:

The reviewer has brought up an excellent point. The curve of $1/C^2_{\text{MoS}_2}$ vs. V_R in Fig. R2 allows us to determine the dopant concentration and built-in potential to be $\sim 3 \times 10^{18} \text{ cm}^{-3}$ and $\sim 0.5 \text{ V}$, in good agreement with results in literature¹⁷, via the slope and the intercept according to¹⁸

$$\frac{1}{C_{\text{MoS}_2}^2} = \frac{2(|\Phi_{bi}| + |V_R|)}{qN_d \epsilon_r \epsilon_0 A^2}. \quad (\text{R3})$$

Then, $W = \sqrt{2\epsilon_r \epsilon_0 (|\Phi_{bi}| + |V_R|) / qN_d}$ gives a depletion width of $\sim 22 \text{ nm}$ under the reverse bias of 0.5 V . Here, we note that due to the much smaller area of MoS_2 compared to that of the electrodes (Fig. S10a), the device capacitance measured includes a parallel parasitic capacitance, which is insensitive to external voltage and hence has no effect on the differential, DLTS signal (please see Fig. S10b). This parasitic capacitance is subtracted to calculate the $1/C^2_{\text{MoS}_2}$ vs. V_R in Fig. R2 (please see details in Fig. S10). Note that the dopant concentration is much higher than free electron density measured by FET in Fig. S9, which is attributed to the trapping of free electrons by DX centers.

Figure R2: $1/C^2_{\text{MoS}_2}$ vs. reverse voltage to characterize the dopant concentration for the device in Fig. 1a at room temperature.

The nearly linear $1/C^2$ vs. V_R curve indicates the roughly uniform distribution of

dopants and approximate step junction profile of space charge density in the surface of MoS₂/Pt Schottky diode^{21,22}, hence confirming the feasibility of using $W = \sqrt{2\varepsilon_r\varepsilon_0(|\Phi_{bi}| + |V_R|)/qN_d}$ to estimate the depletion width.

A more explicit explanation has been added to Fig. S10 in the supplementary information.

Q3: *Also, the same study but as a function of frequency would inform about the charge states. But then a technical issue is raised. What is the oscillation level (V_{osc}) superimposed to the constant V_R , necessary to carry capacitance measurements? In the present case the depletion region seems so tiny (~21 nm) that a very small value of V_{osc} is required to have meaningful data. What is this value?*

Response:

In our case, the oscillation level has an amplitude of ~ 60 mV with a frequency of 1MHz. The overall effect of the alternating voltage can be viewed as a small oscillation of the depletion width around the equilibrium state. Compared to the V_R of 0.5V in the DLTS test, this oscillation level is small enough to ensure only small amount of charges are being charged to or discharged from the parallel depletion capacitor.

The method section has been modified to include this technical parameter.

Q4: *Finally, the depletion is so small that Debye incursion cannot be neglected which should certainly affect the shape of the emission transient from which DLTS data are extracted. A serious deviation from pure exponential is very likely. But not only. Do the authors know the mobility of free carriers. In case of low mobility, multi-trapping process is not impossible which should also affect the shape of the transient. All these aspects would have consequences on data analysis.*

We agree with the reviewer's comment on the influence of Debye incursion on the depletion capacitance. Despite the feasibility of the step junction profile for space charge, more precisely, the free carrier density decreases exponentially within the depletion zone, so a Debye screening length (or Debye tail, Debye incursion) is defined to express the abruptness of the space charge distribution near the edge of the depletion zone, which can be expressed as²¹

$$L_D = \sqrt{\frac{\varepsilon_r\varepsilon_0kT}{q^2N_d}}. \quad (\text{R4})$$

In our case, the high dopant concentration, $N_d \sim 3 \times 10^{18} \text{ cm}^{-3}$ derived via Fig.R2, yields the Debye length of ~ 2 nm at room temperature, which is in good agreement with that in heavily doped traditional semiconductors²¹. The depletion width (~ 22 nm) is more than ten times greater than the Debye length, thereby resulting in the Debye incursion effect to be negligible²¹.

Figure R3: Capacitance transient and its exponential fit in the DLTS signal.

The capacitance transient can be well fitted by the exponential function, in particular for narrow time windows (< 20 ms), thus excluding any considerable effect of Debye incursion on the shape of the DLTS signal, as shown in Fig. R3.

The multi-trapping process usually occurs in amorphous or organic systems with ultralow mobility^{23,24}, such as the hole transport in SiO₂ with the mobility of $\sim 10^{-6}$ cm²/(V·S)²³; whereas, in our case, the mobility of MoS₂ is measured to be ~ 16 cm²/(V·S) as shown in the new Fig.5, so it is reasonable to rule out the possibility of multiple trapping in the TMDs.

We have added a related explanation to Fig. S10.

Q5: *It is not clear whether the condition for an easy analysis of DLTS spectra, that is $N_T/N_D \ll 1$, is fulfilled or not. N_T being the electrically active concentration of vacancies or DX centers, and N_D the free electron concentration if one assumes that all intentional or no intentional dopants are ionized leading to N cited above. The authors should give an estimation of this ratio starting from an estimation of the electronically active concentration of vacancies.*

Response:

Based on the principle of DLTS test⁹, the relationship for the electron trap can be expressed as $N_T/N_D = 2 \times (\Delta C/C)$, so, in our case of MoS₂, N_T/N_D was estimated to be ~ 0.05 from the raw DLTS signal and background capacitance under quiescent reverse bias, verifying the assumption of $N_T/N_D \ll 1$ for the DLTS measurement. Since N_D is obtained to be $\sim 3 \times 10^{18}$ cm⁻³ via CV characterization as in Fig. R2, the electrically active concentration of V_S is then $\sim 3 \times 10^{17}$ cm⁻³, which is smaller than STEM-seen V_S density due to the unknown passivation effect, as discussed in the main text and Fig. S6.

Q6: *The I-V characteristics displayed in Fig. 1e show that at high temperatures the authors must be cautious. The conditions of meaningful DLTS measurements may break down. One of the major assumptions in DLTS is that during the emission process the only carriers present in the depletion region are those coming from the traps. Here, the authors may have to consider those participating in the leakage current coming from all parts of the device.*

Response:

This is an excellent point. We agree with the reviewer that leakage current may lead to invalidity of the depletion capacitance measurement. We draw the equivalent circuit of the Schottky junction in Fig. R4.

Figure R4: Equivalent circuit of the device in Fig. 1a. The yellow region represents the Schottky barrier circuit, showing a depletion capacitance of MoS₂ (C_{MoS_2}) with a parallel leakage resistance (R_L) and series resistance of the non-depleted region (R_S).

When subtracting the parallel stray capacitance, the measured capacitance, C_{device} in the circuit within the yellow shadow (Fig. R4), is correlated to C_{MoS_2} by²¹

$$\frac{C_{\text{MoS}_2}}{C_{\text{device}}} = \left(1 + \frac{R_S}{R_L}\right)^2 + \left(\frac{R_S}{1/\omega C_{\text{MoS}_2}}\right)^2, \quad (\text{R5})$$

where ω is the frequency of a.c. voltage during the capacitance measurement and is 1 MHz in our case. Accurate test of the depletion capacitance and hence the depletion width requires that $R_S \ll R_L$ and $R_S \ll 1/\omega C_{\text{MoS}_2}$, such that the capacitive impedance, C_{MoS_2} , dominates the circuit element^{21,22,25}. The leakage resistance (R_L) and series resistance (R_S) can be approximately estimated by reverse and forward bias current of the Schottky junction to be 80 k Ω and 2.5 k Ω under the reverse bias of 0.2V at 320 K (Fig. 1e), meeting the requirement of $R_S \ll R_L$. Then, given that $C_{\text{MoS}_2} \approx C_{\text{total}} - C_{\text{stray}}$ ($C_{\text{stray}} = 220$ pF, in Fig. S10), $1/\omega C_{\text{MoS}_2} = 77$ k $\Omega \gg R_S$ at 320 K. Therefore, Equation (R5) gives $C_{\text{MoS}_2} \sim C_{\text{device}}$, which confirms the dominance of MoS₂ capacitance and hence the reliable capacitance measurement. We note that the large leakage current under reverse bias, called ‘soft’ reverse characteristics, may be attributed to the tunneling effect or the reduced Schottky barrier height induced by image force lowering, as commonly reported in the Schottky junctions formed by low dimensional materials²⁶⁻²⁸.

Figure S10 is added in the Supplementary to show this discussion in details.

Q7: Fig. 2 and lines 121 to 142 in the manuscript and Fig. S1 discuss the results collected from 4 samples with different alloy compositions. Fig. S1 is specially

confusing as it is not clear which DLTS/CTS spectra corresponds to what alloy composition. On the other hand, if we refer to Fig. 2 and Fig. S1d, the enthalpy of ionization attributed to the vacancy is clearly lower in MoS₂ than in [Mo] 0.3 W_{0.7}S₂. However, the authors claim that the vacancy level follows the CBM for all studied compositions. Do they mean the vacancy level is pinned to CB? Clarifications are needed.

Response:

The labels of the composition are already given above each DLTS/CTS plot.

What we mean in the manuscript is that the V_s level “largely follows” the CBM of the $Mo_{1-x}W_xS_2$ host, rather than exactly pinned to CBM or lining up at a fixed position with respect to the vacuum level. This is because, as shown in the partial density of states plot in Fig. 2d, V_s states originate mostly from the 4d (5d) orbitals of Mo (W) atoms, hence sharing similar orbital composition as the CBM and following its trend with the hosts^{13,29}. Looking more closely, Figure 2d presents that the V_s state is not completely composed of d orbitals of Mo(W) atoms, instead, it contains also 3p orbitals from S atoms, such that its energy level deviates slightly from following the exact CBM of the host.

At the same time, we note that the activation energies determined by DLTS inevitably have error bars, arising from variations of distinct samples, different measurement parameters, and fitting of DLTS features. Therefore, the 0.05 eV activation energy difference between MoS_2 and $Mo_{0.3}W_{0.7}S_2$ is within the range of errors, ± 0.03 eV for MoS_2 and ± 0.02 eV $Mo_{0.3}W_{0.7}S_2$.

Q8: *The deeper level in Fig. S1d reflects clearly the alloy disorder at the atomic level. The peak is wider and asymmetric in contrast with what DLTS or CTS models predict in a homogeneous material. Conventional DLTS is of course unable to sort out the various structures hidden by the broadening and corresponding to various local alloy chemical disorder. Laplace DLTS could help in some restricted conditions. This leads to claim that the energy extracted from the Arrhenius law is an average and the extracted (not measured) capture cross section is in such cases meaningless.*

Response:

We agree with the reviewer’s comments on Laplace DLTS, an isothermal measurement with much higher energy resolution than conventional DLTS, which is unfortunately currently unavailable for us. Even so, the standard DLTS in our experiment is sensitive enough to detect the alloy-disorder-averaged activation energies of V_s states and probe their chemical trends. Finer structures of these features due to alloy disorder, as the reviewer pointed out, would be an interesting topic of research for the future.

Q9: *The part dedicated to PPC results is clearly detailed and seems convincing. However, the link of the emission energy E_e to the barely seen peak B in Fig. 2a is puzzling to say the least. All data resulting from PPC measurement seem to indicate a very significant concentration of the metastable DX center that the authors attribute to PPC signal. At 250K corresponding to the lowest rate window, the peak is so weak that it cannot be linked to PPC, unless the weak signal corresponding to peak B has a*

physical meaning.

Response:

We thank the reviewer's positive evaluation of our PPC results.

The reason for assigning the feature B as DX centers is because the binding energy (E_b) and capture barrier (E_c), as shown in the configurational coordinate diagram in Fig. R1, are estimated to be ~ 0.12 and 0.28 eV, respectively, for MoS₂ by the temperature dependence of conductance and PPC transient test, and their sum is consistent with the energy of feature B, ~ 0.4 eV.

Figure R5: DLTS signal of MoS₂ at various pulse width.

If the filling pulse time is sufficiently long that the DLTS feature is saturated, the height of the DLTS feature represents the occupation of the DX centers (not exact DX center density)³⁰. However, owing to the large capture barrier of ~ 0.2 eV for MoS₂, the DLTS peak does not reach saturation within the pulse width, as shown in Fig. R5, where DX centers feature with the pulse of 400ms is higher than that of 200ms, in contrast to the constant height of V_s feature. Thus, the feature intensity indicates the density of carriers trapped at DX centers during the pulse but underestimates the true density, an effect also observed in traditional III-V semiconductors³¹⁻³³.

Q10: *For the clarity of the paper, it would be useful to discuss briefly the structure of the bandgap (direct vs indirect) for both MoS₂ and WS₂. In relation to this, the behavior of the band gap as a function of alloying would also be worth discussing as it has fundamental impact on the position of levels in the band gap discussed in the manuscript.*

Response:

The direct (monolayer) vs indirect (bulk) transition of bandgap is less related to our work. This is because the very weak interlayer vdW couplings only modulate the band edges (CBM and VBM), hardly affecting the entire band structure¹³. The spatially very localized wavefunctions of deep levels do not hybridize only with these band edges (which is in contrast to shallow defects whose wavefunctions are composed of entirely the band edge states).

Regarding the bandgap as a function of alloying, monolayer $\text{Mo}_x\text{W}_{1-x}\text{S}_2$ alloys have a small band bowing with the minimum bandgap value at $x=0.2$ ³⁴, due to possible band-anticrossing effect (BAC)^{35,36}; whereas, for multilayer ones, the indirect bandgap evolution with the host composition is generally considered as a simple interpolation of the end-point bandgaps.

We have included this discussion in the revised manuscript (Page 14, Paragraph 2).

Q11: *Regarding Note 1 in supporting information*

Although the relation of the emission rate used by the authors (equation (1)) is widely accepted in literature, the activation energy has no physical meaning if not expressed in thermodynamic framework. The appropriate energy is the Gibb's free energy which is split into the enthalpy appearing in the exponential and the entropy which appears in the pre-factor. Two consequences result; (i) the emission enthalpy should be temperature dependent except in the case of a strict pinning to the respective allowed band (here the conduction band). It is however understandable that the narrow temperature range allowed by the accessible rate windows might make such a study difficult. This is why one consider the enthalpy roughly temperature independent.

But, unless the capture cross section is measured, its extraction from the pre-factor, which is got from the intercept at $T=\infty$, should include the entropy factor. Therefore, the extrapolation leads to the product $e^{(\Delta S_n/k)} \sigma_n$ and not σ_n , where k stands for Boltzmann constant and ΔS_n for the emission entropy. In the present context the authors overestimate the capture cross section by a non-negligible factor $e^{(\Delta S_n/k)}$. One may expect the vibrational modes to be more sensitive to trapping – emission in vdW materials than in conventional covalent materials. The authors mention in line 53-54 of the manuscript that stronger lattice relaxations are to be expected in vdW materials. They should thus comment on this with respect to ΔS_n .

If the authors decide to measure the capture cross section they must be aware of non-linearities due to very narrow depletion region for which Debye incursion of free carriers can no longer be neglected.

Response:

We appreciate the reviewer's valuable suggestions on the thermodynamic view of activation energies from DLTS, which could make our interpretation more solid.

The defect energy level in semiconductors is defined as the change of chemical potential due to the formation of a carrier - ionized defect pair^{37,38}. The chemical potential can be thermodynamically expressed as the variation of Gibbs free energy during the capture or emission of an electron at constant pressure and temperature. Thus, based on these definitions, the Arrhenius equation of the thermal emission rate in Eq. (1) can be rewritten as³⁸

$$\frac{e_n}{T^2} = K \sigma_n \exp\left(-\frac{\Delta G(T)}{k_B T}\right), \quad (\text{R6})$$

where $\Delta G(T) = |E_{CB} - E_i|$ and is the activation energy for electron emission from the deep state to the conduction band edge. At the same time, the Gibbs free energy is defined by the thermodynamic identity as $\Delta G(T) = \Delta H - T\Delta S$, where ΔH and ΔS represent the changes in enthalpy and entropy, respectively. Therefore, Equation (R6) becomes²¹

$$\frac{e_n}{T^2} = K[\exp\left(\frac{\Delta S}{k_B}\right)\sigma_n]\exp\left(-\frac{\Delta H}{k_B T}\right), \quad (\text{R7})$$

and hence the slope of the Arrhenius plot via Eq. (1) yields an average of enthalpy change over the temperature range of this plot, considering the generally weak temperature dependence of ΔH . The difference between ΔG and ΔH mainly arises from lattice vibrational contribution to ΔS due to the coupling of occupied deep states to the lattice, so, usually, it is negligible when electrons are excited from the traps to conduction band without invoking a change in the bonding configuration ($\Delta S \sim 0$)³⁹. Thus, in this study, it is reasonable to consider the measured Arrhenius slope from DLTS as the activation energy for V_s states, because our DFT calculations do not observe lattice relaxation or entropy change during the transfer of electrons between V_s and conduction band edge.

With regard to DX centers, most of previous studies on III-V semiconductors neglected the difference between ΔG and ΔH as well^{19,20,31,40-43}, despite the occurrence of lattice relaxation when switching with the electron-donating state. Hence in this study we do not consider this difference for DX centers in vdW crystals.

After obtaining the linear $1/C^2$ vs. V curve (Fig. R2) and the Debye length (response to Q4), the main text and note 1 in SI present the extraction of capture cross section, σ_n , of V_s deep state via the intercept of the Arrhenius plot in Fig. 2b, but we note that, based on Eq. (R7), this intercept represents the product $\exp\left(\frac{\Delta S}{k_B}\right)\sigma_n$, rather than σ_n . Experimentally, one can measure the latter using the diode shorting-circuiting technique^{44,45}, thereby determining the prefactor, ΔS , by temperature-dependent Gibbs free energy ($\Delta G = \Delta H - T\Delta S$).

We have added the thermodynamic explanation in the revised manuscript (Page 5, Paragraph 1) and note 5 in the Supplementary information to make the principle of DLTS clearer.

Point-by-point response to Reviewer #3's comments:

Reviewer #3 (Remarks to the Author):

Overall comment: *Defects are important for atomically thin 2D materials. It is a topic that is not fully investigated. However, the authors are investigating the bulk layered materials instead of single-layer atomically thin 2D materials. More importantly, the reviewer doesn't believe that the model of DX defects is realistic unless the authors fully address the following points:*

Response:

We thank the reviewer for reviewing our manuscript and recognizing the importance of defect studies in 2D materials. The reasons we measure bulk layered materials instead of monolayers are already given in response to Q4 of Reviewer#1.

We reply to the reviewer's questions about the DX center model below.

Q1: *If the model is real, the authors should perform photo Hall measurements to verify whether it is the electrons contributing to the persistent photocurrent.*

Response:

This is an excellent suggestion. Photo-Hall effect would directly measure the carrier density (rather than conductivity) before and after the carrier de-trapping. The transient persistent photocurrent we performed (Fig. 4a) is generally considered to have a similar relaxation trend with free carrier density in group III-V semiconductor study, and hence can be used to extract the delay time constant via an Arrhenius plot, because changes in carrier mobility are typically much weaker compared to the variation of conductance⁴⁶⁻⁵⁰, and at the same time, conductivity (σ) is the product of carrier density (n) and mobility (μ): $\sigma = n \cdot e \cdot \mu$.

Following the reviewer's suggestion, in order to thoroughly verify the dominant role of free carrier trapping/de-trapping in the PPC effect (rather than a mobility variation effect), a multilayer MoS₂ field-effect transistor (FET) was made to determine the carrier mobility before and after the light illumination, which would yield the same information with photo Hall measurements. Figure R6a shows that a multilayer MoS₂ is deposited on a heavily doped silicon substrate with an insulator layer of SiO₂ (300nm), acting as a back gate (V_G). According to the data in Fig. R6b, we can extract the low-field field-effect mobility to be $\sim 16 \text{ cm}^2/(\text{V}\cdot\text{S})$ for MoS₂ (consistent with the previous reports⁵¹), based on the expression $\mu = [dI_{DS}/dV_{Gate}] \times [L/WCV_{DS}]$ ⁵², where W and L are the width and length of the channel, respectively, C is the capacitance of SiO₂ insulator, and V_{DS} represents the source-drain voltage in the MoS₂ FET. The $I_{DS} - V_G$ curves exhibit the same slope before and after the illumination, as shown in the solid lines in Fig. R6b, indicating a constant electron mobility regardless of electron density or light illumination. Hence these results confirm that the time dependence and kinetics of conductivity in PPC (Fig.4a) genuinely reflects the time dependence of free electron density in the material, validating the extraction of time constant via Eq. (2) for the DX center model.

Figure R6: Transient mobility of MoS₂ before and after illumination. a, Schematic of a multilayer MoS₂ field-effect transistors (FET) from the cross-sectional view. **b,** Transient transfer characteristic for the FET with 25 mV applied bias voltage V_{DS} before (hollow symbols) and after (solid symbols, at different times after turned off) light illumination at 300 K. The back gate voltage, V_G , is applied to the substrate. The solid lines represent the slopes of $I_{DS} - V_G$ curves, indicating a nearly-constant electron mobility of this device regardless of the electron density. Therefore, kinetics of free electrons follows the kinetics of the material's conductivity, and conductivity variation directly reflects the free electrons trapping/de-trapping.

The above results have been added in Fig. 5 and related discussion part of the main text.

Q2: Based on the model, how to explain the long rise time in Fig3.b (blue line)?

Response:

This could be attributed to the fact that, as a metastable defect, DX centers are able to switch with shallow dopants by capturing or emitting free carriers, but this process needs time to reach equilibrium, depending on the height of capture barrier. The long rise time in PPC is a common feature for DX centers, as usually also observed in traditional semiconductors^{49,50}.

Q3: The DX concentration is unlikely high enough to induce the high photoconductivity. The authors can roughly estimate the concentration of DX defects from the photoconductivity.

Response:

We thank the reviewer's valuable suggestions on estimating the DX centers density from the change of photoconductivity. By the $I_{DS}-V_G$ curve of FETs, the free carrier concentration of MoS₂ is measured to be 4×10^{17} and $2 \times 10^{18} \text{ cm}^{-3}$ before and after illumination at 300 K (see details in Fig. R6b and Fig. S9), thereby leading to the rough DX center concentration of $\sim 1.6 \times 10^{18} \text{ cm}^{-3}$, in good agreement with the dopant concentration determined by capacitance measurement (Fig. S10). However, even though this approach was used to estimate DX center density in traditional III-V semiconductors⁵³, we have to note that the difference of free carriers between before and after illumination is the change of carrier occupation of DX centers, which is on

the same order of the total DX centers density when the PPC effect is strong³⁰.

Q4: *It is unclear the wavelength of excitation light used in the experiments. The author should illuminate the sample with photon energy lower than the bandgap so that only electrons in the DX defects can be excited to the conduction band.*

Response:

We used white light to illuminate the samples (please find the detailed parameter in the Method section). Although white light excites electron-hole pairs across the bandgap together with the emission of electrons from DX centers, photo-generated electrons and holes from the valence band do not affect the DX center model, because, after the light is off, the recombination of electron-hole pairs occurs on the order of nanoseconds⁵⁴, much faster than the relaxation of electrons to DX centers (milliseconds). This fast recombination process corresponds to the abrupt drop of the current transient in Fig. 3b, which was subtracted off when fitting the delay time constant of PPC. Moreover, the threshold of phonon energy to excite electrons from the DX center is probably also greater than the bandgap, as shown by the vertical arrow in Fig.3c(2), and a sub-bandgap light illumination would not ionize the DX centers. Therefore, the photon-generated electron-hole pair by white light will not affect the capture of electrons by DX centers after the illumination, as verified as well by previous studies in traditional semiconductors^{30,46,49}.

Q5: *The authors should thoroughly investigate the material itself. It is unclear whether the semiconductor is n-type or p-type and what is the conductivity.*

Response:

The samples have been confirmed to be n-type semiconductors by three experiments in the main text, including the rectifying IV and CV characteristics of the Schottky junction (Fig. 1 e and f) and $I_{DS}-V_G$ curve of FETs (Fig. 5b).

Q6: *Lastly, the authors actually investigate the bulk layered materials. The bandgap and band structure will change as the materials become atomically thin. The information on the defect energy levels found in this manuscript may not be applicable to the defects in atomically thin 2D materials.*

Response:

The reviewer has brought up a good point, which is the same as Q4 of Reviewer#1. Despite that the bandgap changes from indirect in multilayer transition metal dichalcogenides (TMDs) to direct in monolayer ones^{11,12}, physical properties of V_S defects and DX centers discovered here in multilayer vdW semiconductors are likely applicable to monolayer and few-layer samples. This is because the very weak interlayer vdW couplings only modulate the band edges, hardly affecting the entire band structure¹³. On the other hand, the spatially strongly localized wavefunctions of deep levels do not only hybridize with the conduction or valence band edges (in contrast to shallow defects whose wavefunction consists entirely of the band edge states). For example, for monolayer MoS₂, first-principles calculations predicted that the V_S deep level lies about 0.5 eV below the CBM at the K point of Brillouin Zone^{2,6-8,14}, which is in good agreement with the 0.27 eV below the CBM at the Q point of

multilayer MoS₂ determined in this study, considering the 0.2 eV CBM band offset between monolayer and multilayer samples^{15,16}. It is therefore reasonable to expect the properties of native defects attained in this study to be applicable to monolayer ones.

DLTS detects the capacitance variation within a given time window across the sample thickness driven by bias voltage, so ultrathin TMDs (such as monolayers) would introduce large high leakage current and hence result in inability or unreliability of capacitance measurements. Meanwhile, the depletion capacitance of Schottky junction requires the sample to be thicker than the depletion width, ~ 20 nm (Fig. S10) for MoS₂, so multilayer TMDs (> 50 nm) were studied in this manuscript.

We have included this discussion in the revised manuscript (Page14, Paragraph 2).

References of Response

- 1 Barja, S. *et al.* Identifying substitutional oxygen as a prolific point defect in monolayer transition metal dichalcogenides. *Nature communications* **10**, 1-8 (2019).
- 2 Vancsó, P. *et al.* The intrinsic defect structure of exfoliated MoS₂ single layers revealed by Scanning Tunneling Microscopy. *Scientific reports* **6**, 29726 (2016).
- 3 Addou, R., Colombo, L. & Wallace, R. M. Surface defects on natural MoS₂. *ACS applied materials & interfaces* **7**, 11921-11929 (2015).
- 4 Liu, X., Balla, I., Bergeron, H. & Hersam, M. C. Point defects and grain boundaries in rotationally commensurate MoS₂ on epitaxial graphene. *The Journal of Physical Chemistry C* **120**, 20798-20805 (2016).
- 5 Jeong, T. Y. *et al.* Spectroscopic studies of atomic defects and bandgap renormalization in semiconducting monolayer transition metal dichalcogenides. *Nature communications* **10**, 1-10 (2019).
- 6 Qiu, H. *et al.* Hopping transport through defect-induced localized states in molybdenum disulphide. *Nature communications* **4**, 1-6 (2013).
- 7 Hong, J. *et al.* Exploring atomic defects in molybdenum disulphide monolayers. *Nature communications* **6**, 1-8 (2015).
- 8 Komsa, H.-P. & Krasheninnikov, A. V. Native defects in bulk and monolayer MoS₂ from first principles. *Physical Review B* **91**, 125304 (2015).
- 9 Lang, D. Deep - level transient spectroscopy: A new method to characterize traps in semiconductors. *Journal of applied physics* **45**, 3023-3032 (1974).
- 10 McCluskey, M. D. & Haller, E. E. *Dopants and defects in semiconductors*. (CRC press, 2018).
- 11 Splendiani, A. *et al.* Emerging photoluminescence in monolayer MoS₂. *Nano letters* **10**, 1271-1275 (2010).
- 12 Mak, K. F., Lee, C., Hone, J., Shan, J. & Heinz, T. F. Atomically thin MoS₂: a new direct-gap semiconductor. *Physical review letters* **105**, 136805 (2010).
- 13 Ci, P. *et al.* Quantifying van der Waals interactions in layered transition metal dichalcogenides from pressure-enhanced valence band splitting. *Nano letters* **17**, 4982-4988 (2017).
- 14 Chen, Y. *et al.* Tuning electronic structure of single layer MoS₂ through defect and interface engineering. *ACS nano* **12**, 2569-2579 (2018).
- 15 Liu, G.-B., Xiao, D., Yao, Y., Xu, X. & Yao, W. Electronic structures and theoretical modelling of two-dimensional group-VIB transition metal dichalcogenides. *Chemical Society Reviews* **44**, 2643-2663 (2015).
- 16 Guo, Y. & Robertson, J. Band engineering in transition metal dichalcogenides: Stacked versus lateral heterostructures. *Applied Physics Letters* **108**, 233104 (2016).
- 17 Liu, Y. *et al.* Approaching the Schottky–Mott limit in van der Waals metal–semiconductor junctions. *Nature* **557**, 696-700 (2018).
- 18 Hu, C. *Modern semiconductor devices for integrated circuits*. Vol. 2 (Prentice Hall Upper Saddle River, New Jersey, 2010).
- 19 Mooney, P., Theis, T. & Wright, S. Effect of local alloy disorder on emission kinetics of deep donors (DX centers) in Al_xGa_{1-x}As of low Al content. *Applied physics letters* **53**, 2546-2548 (1988).
- 20 Theis, T., Mooney, P. & Wright, S. Electron Localization by a Metastable Donor Level in n–GaAs: A New Mechanism Limiting the Free-Carrier Density. *Physical review letters* **60**, 361 (1988).
- 21 Blood, P. & Orton, J. W. *The electrical characterization of semiconductors*:

- majority carriers and electron states. Vol. 2 (Academic press London, 1992).
- 22 Pierret, R. F. *Semiconductor device fundamentals*. (Pearson Education India, 1996).
- 23 Curtis Jr, O. & Srour, J. The multiple - trapping model and hole transport in SiO₂. *Journal of Applied Physics* **48**, 3819-3828 (1977).
- 24 Montero, J. M. & Bisquert, J. Trap origin of field-dependent mobility of the carrier transport in organic layers. *Solid-state electronics* **55**, 1-4 (2011).
- 25 Goodman, A. M. Metal—Semiconductor barrier height measurement by the differential capacitance method—One carrier system. *Journal of Applied Physics* **34**, 329-338 (1963).
- 26 Kumar, S. *et al.* Influence of barrier inhomogeneities on transport properties of Pt/MoS₂ Schottky barrier junction. *Journal of Alloys and Compounds* **797**, 582-588 (2019).
- 27 Rhoderick, E. H. & Rhoderick, E. *Metal-semiconductor contacts*. (Clarendon Press Oxford, 1978).
- 28 Chen, C.-C., Aykol, M., Chang, C.-C., Levi, A. & Cronin, S. B. Graphene-silicon Schottky diodes. *Nano letters* **11**, 1863-1867 (2011).
- 29 Kang, J., Tongay, S., Zhou, J., Li, J. & Wu, J. Band offsets and heterostructures of two-dimensional semiconductors. *Applied Physics Letters* **102**, 012111 (2013).
- 30 Mooney, P. Deep donor levels (DX centers) in III - V semiconductors. *Journal of Applied Physics* **67**, R1-R26 (1990).
- 31 Kasu, M., Fujita, S. & Sasaki, A. Observation and characterization of deep donor centers (DX centers) in Si - doped AlAs. *Journal of applied physics* **66**, 3042-3046 (1989).
- 32 Mizuta, M., Tachikawa, M., Kukimoto, H. & Minomura, S. Direct evidence for the DX center being a substitutional donor in AlGaAs alloy system. *Japan. J. Appl. Phys.* **24**, L143-L146 (1985).
- 33 Tachikawa, M., Mizuta, M. & Kukimoto, H. DX Deep Centers in Al_xGa_{1-x}As Grown by Liquid-Phase Epitaxy. *Japanese Journal of Applied Physics* **23**, 1594 (1984).
- 34 Chen, Y. *et al.* Tunable band gap photoluminescence from atomically thin transition-metal dichalcogenide alloys. *Acs Nano* **7**, 4610-4616 (2013).
- 35 Shan, W. *et al.* Band anticrossing in GaInNAs alloys. *Physical Review Letters* **82**, 1221 (1999).
- 36 Wu, J., Shan, W. & Walukiewicz, W. Band anticrossing in highly mismatched III–V semiconductor alloys. *Semiconductor Science and Technology* **17**, 860 (2002).
- 37 Van Vechten, J. & Thurmond, C. Entropy of ionization and temperature variation of ionization levels of defects in semiconductors. *Physical Review B* **14**, 3539 (1976).
- 38 Thurmond, C. The standard thermodynamic functions for the formation of electrons and holes in Ge, Si, GaAs, and GaP. *Journal of the Electrochemical Society* **122**, 1133 (1975).
- 39 Almbladh, C.-O. & Rees, G. Statistical mechanics of electronic energy levels in semiconductors. *Solid State Communications* **41**, 173-176 (1982).
- 40 Criado, J., Gomez, A., Munoz, E. & Calleja, E. Deep level transient spectroscopy signature analysis of DX centers in AlGaAs and GaAsP. *Applied physics letters* **49**, 1790-1792 (1986).
- 41 Calleja, E., Gomez, A., Munoz, E. & Camara, P. Fine structure of the alloy -

- broadened thermal emission spectra from DX centers in GaAlAs. *Applied physics letters* **52**, 1877-1879 (1988).
- 42 Calleja, E., Gomez, A. & Munoz, E. Direct evidence of the DX center link to the L - conduction - band minimum in GaAlAs. *Applied physics letters* **52**, 383-385 (1988).
- 43 Kumagai, O., Kawai, H., Mori, Y. & Kaneko, K. Chemical trends in the activation energies of DX centers. *Applied physics letters* **45**, 1322-1323 (1984).
- 44 Brotherton, S. & Lowther, J. Electron and hole capture at Au and Pt centers in silicon. *Physical Review Letters* **44**, 606 (1980).
- 45 Brotherton, S. & Bicknell, J. The electron capture cross section and energy level of the gold acceptor center in silicon. *Journal of Applied Physics* **49**, 667-671 (1978).
- 46 Lin, J., Dissanayake, A., Brown, G. & Jiang, H. Relaxation of persistent photoconductivity in Al_{0.3}Ga_{0.7}As. *Physical Review B* **42**, 5855 (1990).
- 47 Fujisawa, T., Krištofik, J., Yoshino, J. & Kukimoto, H. Metastable behavior of the DX center in Si-doped GaAs. *Japanese journal of applied physics* **27**, L2373 (1988).
- 48 Tachikawa, M. *et al.* Observation of the persistent photoconductivity due to the DX center in GaAs under hydrostatic pressure. *Japanese journal of applied physics* **24**, L893 (1985).
- 49 Li, J. *et al.* Nature of Mg impurities in GaN. *Applied physics letters* **69**, 1474-1476 (1996).
- 50 McCluskey, M. *et al.* Metastability of oxygen donors in AlGaIn. *Physical Review Letters* **80**, 4008 (1998).
- 51 Radisavljevic, B. & Kis, A. Mobility engineering and a metal-insulator transition in monolayer MoS₂. *Nature materials* **12**, 815-820 (2013).
- 52 Radisavljevic, B., Radenovic, A., Brivio, J., Giacometti, V. & Kis, A. Single-layer MoS₂ transistors. *Nature nanotechnology* **6**, 147 (2011).
- 53 Oelgart, G. *et al.* Hall effect, photoluminescence and DLTS investigation of the DX centre in AlGaAs. *Semiconductor science and technology* **5**, 894 (1990).
- 54 Amani, M. *et al.* Near-unity photoluminescence quantum yield in MoS₂. *Science* **350**, 1065-1068 (2015).

REVIEWER COMMENTS

Reviewer #1 (Remarks to the Author):

After reading the revised manuscript and the reply to all the reviewers comments, I found that the authors have extensively addressed all my criticism and implemented all the changes. I am happy to recommend this paper for publication.

Reviewer #2 (Remarks to the Author):

The authors have fully and in detail responded to all my questions and remarks. The answers are convincing and the authors mentioned clearly the changes made in the manuscript and supplementary information. Therefore, I consider that the manuscript is now much more consistent, appealing to new ideas and hence deserves publication in Nature communication
Mesli

Reviewer #3 (Remarks to the Author):

The authors have addressed most of the questions that the reviewer asked previously. Although convincing in many aspects, there is a possibility that the observed persistent photocurrent originates from the trapping effect from minority trap states, which are not visible in DLTS. The reviewer suggests the authors do the following two experiments:

- 1) One easy experiment that authors can do is to measure the spectral responses of these devices to check whether the energy to excite the charge carriers from DX centers is consistent with the picture presented in the manuscript. If the photoresponses come from the trapping effect of minority trap states, then majority carriers will be left in the conduction band to contribute to the photocurrent. In this case, high photoresponses (high photo gain) are often observed when the photon energy is larger than the bandgap to create electron-hole pairs.
- 2) The authors should seriously analyze the fast transient responses in Fig.3 ("the abrupt drop of the current transient"). The photoresponses of many semiconductors including 2D materials often have a fast and a slow component. The fast component comes from the surface depletion region narrowing which is often on an order of 0.1ms to 10ms. The slow component originates from the emission or capture process of trap states, which can be very long. If the fast component is significantly smaller than 0.1ms, then we can validate the DX model and exclude the possibility of minority trap states creating the observed photoresponses in the manuscript.

Reviewer #1 (Remarks to the Author):

After reading the revised manuscript and the reply to all the reviewers comments, I found that the authors have extensively addressed all my criticism and implemented all the changes.

I am happy to recommend this paper for publication.

Reviewer #2 (Remarks to the Author):

The authors have fully and in detail responded to all my questions and remarks. The answers are convincing and the authors mentioned clearly the changes made in the manuscript and supplementary information.

Therefore, I consider that the manuscript is now much more consistent, appealing to new ideas and hence deserves publication in Nature communication

Mesli

Response:

We thank the reviewers #1 and #2 for recommendation of this manuscript for publication.

Point-by-point response to Reviewer #3's comments:

Overall comment: *The authors have addressed most of the questions that the reviewer asked previously. Although convincing in many aspects, there is a possibility that the observed persistent photocurrent originates from the trapping effect from minority trap states, which are not visible in DLTS. The reviewer suggests the authors do the following two experiments:*

Response:

We thank the reviewer for re-reviewing our manuscript and considering our results convincing. In fact, one can exclude the suggested possibility of minority traps to induce persistent photocurrent (PPC) in this study, because DLTS is capable of detecting and displaying minority deep states at high sensitivity as negative-valued features¹; however, our DLTS spectrum in Fig. 2a does not show negative-valued signals, ruling out the existence of minority deep traps. In the meantime, due to the rapid recombination rate of regular trap-assisted recombination (on the order of ns or ps)², the presumed minority deep levels could not explain the PPC effect with such a long relaxation time of $\sim 10^5$ s at room temperature (Fig. 4). Please find our detailed explanations and point-to-point replies as follows.

Q1: *One easy experiment that authors can do is to measure the spectral responses of these devices to check whether the energy to excite the charge carriers from DX centers is consistent with the picture presented in the manuscript. If the photoresponses come from the trapping effect of minority trap states, then majority carriers will be left in the conduction band to contribute to the photocurrent. In this case, high photoresponses*

(high photo gain) are often observed when the photon energy is larger than the bandgap to create electron-hole pairs.

Response:

Optical threshold (E_{opt}) is typically not used as a characteristic energy to explore the existence of DX centers. The capture barrier (E_c), binding energy (E_b), and emission barrier (E_e) in Fig. 4c and Fig. R1 are the characteristic energies of DX centers; that is, $E_e = E_b + E_c$, as verified in this study and relevant reports of III-V semiconductors³⁻⁷.

It is also possible to exclude the possibility of minority trapping even without the determination of the optical threshold. As mentioned, DLTS is a powerful method of displaying both majority and minority deep states at high sensitivity as positive and negative peaks, respectively¹. However, no negative signals in capacitance change was observed in our DLTS spectrum in Fig. 2a, ruling out the possibility of high concentration minority-trapping deep levels in MoS₂. Meanwhile, Shockley-Read-Hall recombination, including the trapping of holes as proposed by the reviewer, usually occurs within a very short lifetime (on the order of ns or ps)², and cannot generate persistent photoconductivity on the order of tens of hours as we observed.

The mechanism of minority trapping, as proposed by the reviewer, was indeed widely reported to cause PPC effect in ZnO nanowires⁸⁻⁹, but that principle is not applicable for our case. For ZnO nanowires, under illumination, adsorbed oxygen ions on the surface can trap photogenerated holes to produce oxygen molecules, which are subsequently desorbed from the surface with the process of $O_2^- + h^+ \rightarrow O_2$ (gas). After turning off the light, the slow adsorption of O₂ gas by surface to drain excess electrons results in the PPC effect⁸⁻⁹. However, in our case, all tests were completed in a high vacuum ($\sim 10^{-6}$ torr), hence preventing interactions with ions from the environments and excluding the possibility of the minority trapping mechanism.

Figure R1. Configurational coordinate diagram of DX centers.

To sum up, an optical spectral measurement would only give the energy needed to vertically excite the trapped electrons to the conduction band energy (E_{opt} in Fig.R1), which plays no role here in the electronic behavior of DX centers (unlike E_e). At the same time, the process of minority trapping is ruled out, and is not applicable to explain the PPC in our case.

Please find our added description of the majority carrier traps from the DLTS spectrum in the manuscript (Page 5 and Paragraph 1).

Q2: *The authors should seriously analyze the fast transient responses in Fig.3 ("the abrupt drop of the current transient"). The photoresponses of many semiconductors including 2D materials often have a fast and a slow component. The fast component comes from the surface depletion region narrowing which is often on an order of 0.1ms to 10ms. The slow component originates from the emission or capture process of trap states, which can be very long. If the fast component is significantly smaller than 0.1ms, then we can validate the DX model and exclude the possibility of minority trap states creating the observed photoresponses in the manuscript.*

Response:

At the moment the illumination is terminated, the abrupt drop of the current in Fig. 3b shows an obvious discontinuity from the dark current transient, which has been widely observed in DX centers in literature, and attributed to recombination of photo-excited electron-hole pairs at a time constant on the order of nano-seconds for MoS₂^{3, 10-13}.

We agree that, in many semiconductors, the photocurrent transient contains a fast and a slow component, in particular for, for example, ZnO nanowires⁸⁻⁹. In the dark, the oxygen molecules can be absorbed by ZnO surface and hence form surface depletion region to reduce the current; after exposure to light, the charged oxygen ions are desorbed causing narrowing of the surface depletion region, associated with capture/emission of carriers by deep levels, so the photocurrent transient is composed of two processes and hence can be well fitted by a double-exponential function⁸⁻⁹. However, as discussed in Q1, the mechanism of PPC in ZnO does not work for our vdW crystals, and the double-exponential function is unable to form a converged fit of photocurrent in Fig. 4a.

1. Lang, D., Deep - level transient spectroscopy: A new method to characterize traps in semiconductors. *Journal of applied physics* **1974**, 45 (7), 3023-3032.
2. Yu, P. Y.; Cardona, M., *Fundamentals of semiconductors: physics and materials properties*. Springer: 1996.
3. Li, J.; Lin, J.; Jiang, H.; Salvador, A.; Botchkarev, A.; Morkoc, H., Nature of Mg impurities in GaN. *Applied physics letters* **1996**, 69 (10), 1474-1476.
4. Lin, J.; Dissanayake, A.; Brown, G.; Jiang, H., Relaxation of persistent photoconductivity in Al 0.3 Ga 0.7 As. *Physical Review B* **1990**, 42 (9), 5855.
5. Chand, N.; Henderson, T.; Klem, J.; Masselink, W. T.; Fischer, R.; Chang, Y.-C.; Morkoç, H., Comprehensive analysis of Si-doped Al x Ga 1- x As (x= 0 to 1): Theory and experiments. *Physical Review B* **1984**, 30 (8), 4481.
6. Kasu, M.; Fujita, S.; Sasaki, A., Observation and characterization of deep donor centers (DX centers) in Si - doped AlAs. *Journal of applied physics* **1989**, 66 (7), 3042-3046.
7. Kumagai, O.; Kawai, H.; Mori, Y.; Kaneko, K., Chemical trends in the activation energies of DX centers. *Applied physics letters* **1984**, 45 (12), 1322-1323.
8. Mallampati, B.; Nair, S.; Ruda, H.; Philipose, U., Role of surface in high photoconductive gain measured in ZnO nanowire-based photodetector. *Journal of Nanoparticle Research* **2015**, 17 (4), 176.
9. Bao, J.; Shalish, I.; Su, Z.; Gurwitz, R.; Capasso, F.; Wang, X.; Ren, Z., Photoinduced oxygen release and persistent photoconductivity in ZnO nanowires. *Nanoscale research letters*

2011, 6 (1), 1-7.

10. Wu, Y.-C.; Liu, C.-H.; Chen, S.-Y.; Shih, F.-Y.; Ho, P.-H.; Chen, C.-W.; Liang, C.-T.; Wang, W.-H., Extrinsic origin of persistent photoconductivity in monolayer MoS₂ field effect transistors. *Scientific reports* **2015**, 5, 11472.

11. Amani, M.; Lien, D.-H.; Kiriya, D.; Xiao, J.; Azcatl, A.; Noh, J.; Madhvapathy, S. R.; Addou, R.; Santosh, K.; Dubey, M., Near-unity photoluminescence quantum yield in MoS₂. *Science* **2015**, 350 (6264), 1065-1068.

12. McCluskey, M.; Johnson, N.; Van de Walle, C. G.; Bour, D.; Kneissl, M.; Walukiewicz, W., Metastability of oxygen donors in AlGaN. *Physical Review Letters* **1998**, 80 (18), 4008.

13. Zhang, W.; Huang, J. K.; Chen, C. H.; Chang, Y. H.; Cheng, Y. J.; Li, L. J., High - gain phototransistors based on a CVD MoS₂ monolayer. *Advanced materials* **2013**, 25 (25), 3456-3461.

REVIEWER COMMENTS

Reviewer #3 (Remarks to the Author):

The authors' responses are not very convincing to the previous questions.

First, DLTS indeed can detect minority trap states. But the high photoresponses from nanomaterials do not need a high concentration of minority trap states, because of the existence of surface depletion region that will cause high photogain as reported in the following papers: J. Phys. Chem. C 2018, 122, 15, 8564–8572; ACS Nano 14 (2020) 3405-3413.

Second, when the authors attributed the abrupt drop in photocurrent to the minority carrier recombination, the authors cited a few papers claiming the drop is on nanoseconds. But none of the papers actually measured the time scale of this drop in photocurrent. This drop is unlikely on nanosecond scale because there is a paper recently published which actually measured the time scale of this abrupt drop in photocurrent of In₂Se₃ nanosheets (see link:

<https://doi.org/10.1088/1361-6528/abac7e>)

There is also some other evidence showing that this abrupt drop cannot be the minority recombination process. For example, in the reference 13 (Advanced materials 2013, 25 (25), 3456-

3461.) that the authors provides in the last response letter, Fig.4d shows that the photocurrent is independent of temperature when $T < 200\text{K}$. The normal SRH model cannot explain this phenomenon because the SRH process is highly dependent on temperature (<https://www.nrel.gov/docs/fy99osti/25482.pdf>).

Lastly, the reviewer acknowledges that it is unfair to ask the authors to adopt one model and reject others in this manuscript. But the authors should seriously discuss and acknowledge the possibility.

REVIEWER COMMENTS

All other reviewers recommended our work for publication in Nature Commun.

Reviewer #3 (Remarks to the Author):

We thank the Reviewer #3 for his/her comments!

Q1: *The authors' responses are not very convincing to the previous questions.*

First, DLTS indeed can detect minority trap states. But the high photoresponses from nanomaterials do not need a high concentration of minority trap states, because of the existence of surface depletion region that will cause high photogain as reported in the following papers: J. Phys. Chem. C 2018, 122, 15, 8564–8572; ACS Nano 14 (2020) 3405-3413.

Response:

We thank the reviewer for sharing the two works in literature. The surface depletion model, in the two cases provided by the reviewer, is based on the presence of oxide layer on the interface with Ge nanowires (*JPCC* paper, as shown in Fig. R1a), and Si nanowires (*ACS Nano* paper)^{1,2}, respectively. However, in our study, mechanically exfoliated, high-quality, crystalline MoS₂ did not possess oxide or impurity layer, confirmed by our scanning transmission electron microscopy (STEM image, Fig. 1b), hence excluding the influence from any undesired interface of imperfections. In fact, free of surface layer or dangling bonds is precisely the key merit of these van der Waals crystals in investigation for potentially revolutionizing electronics.

Figure R1. (a) Schematic of the photoconduction mechanism in Ge NWs with high photogain (*J. Phys. Chem. C*, 2018, 122, 15, 8564)². (b) Transient photocurrent response of Si NWs with different surface passivation (*Nanoscale*, 2018, 10, 82)³.

More importantly, even though the surface depletion model successfully explained the high photogain (or responsivity) of photoconductivity in these two papers^{1,2}, neither of them exhibits PERSISTENT photoconductivity (PPC) effect^{1,2}, and is therefore unable to explain our experimental results. After reviewing more literature, we found that the group of that *ACS Nano* paper also reported transient photoconductivity of Si NWs in response to surface passivation (Fig. R1b)³. Using DPP (Diethyl 1-propylphosphonate)

to modify the surface, higher trap density at the electron quasi Fermi level results in a longer carrier lifetime (\sim tens of seconds) than dry oxide passivated one³. This proves the significant role of passivation layers on the “surface depletion model” in Si or Ge NWs, rather than their intrinsic properties. In contrast, our long and strong PPC effect ($\sim 10^5$ sec) in atomically surface clean MoS₂ is unlikely to originate from such surface modification mechanism.

We emphasize that our samples were carefully exfoliated from bulk MoS₂ crystals of the highest quality as proven by STEM imaging, and caution was taken in our device fabrication (e.g., avoiding conventional, hot formation of Schottky contact) to minimize any potential surface damage.

Q2: Second, when the authors attributed the abrupt drop in photocurrent to the minority carrier recombination, the authors cited a few papers claiming the drop is on nanoseconds. But none of the papers actually measured the time scale of this drop in photocurrent. This drop is unlikely on nanosecond scale because there is a paper recently published which actually measured the time scale of this abrupt drop in photocurrent of In₂Se₃ nanosheets (see link: <https://doi.org/10.1088/1361-6528/abac7e>)

Response:

We claim the abrupt drop of photocurrent after illumination to be the band-to-band recombination, because this pre-PPC drop has been widely observed and accepted in DX-center induced photoconductivity response, reported in traditional III-V semiconductors^{4,5}. For instance, due to the DX centers, GaAs thin film exhibits strong PPC effect under hydrostatic pressure as shown in Fig. R2⁴, where the open circle and filled circle are photoconductance under and immediately after the light illumination, respectively, showing an abrupt drop of the conductance at the moment when illumination is turned off. Such abrupt drop was also observed in our MoS₂ samples (Fig. 3a and b), so we believe it arises also from the recombination process of photo-excited electron-hole pairs.

Figure R2. Temperature dependent conductivity of GaAs before (solid lines) and after (dotted lines) exposure to light under hydrostatic pressure (30 kbar), where

the open circle and filled circle corresponds to photoconductance under and immediately after light illumination, respectively (*Japanese journal of applied physics, 1985, 24.11A, L893*)⁴. Our PPC data in MoS₂ shows very similar behavior.

Even though there is no direct measurement of the carrier lifetime of band-to-band transition in MoS₂ by transient photocurrent response (due to, mainly, difficulty in nano-seconds electrical measurements), this lifetime has been directly measured optically from the decay of time-resolved photoluminescence or reflection spectrum, and found to be on the order of nanoseconds in literatures^{6,7}.

We thank the reviewer for sharing another work about the surface depletion model in α -In₂Se₃ nanosheets (*Nanotechnology, 2020, 31.46, 465201*)⁸, as shown in Fig. R3. In this literature, a very weak PPC effect was observed in α -In₂Se₃ nanosheets with two components, a rapid decay (time constant, τ_{d1} , of $\sim 600 \mu\text{s}$) and a slow decay ($\tau_{d2} \sim 4 \text{ s}$) in Fig. R3a⁸. Akin to Ge and Si nanowires in Fig. R1^{2,3}, for α -In₂Se₃ nanosheets, the validity of the surface depletion model has to be associated with the existence of impurities on the surface; specifically, the appearance of their PPC effect was attributed to the capture of the photo-excited holes by the adsorbents O₂⁻ on the surface (Fig. R3b). However, our STEM image in Fig. 1b did not detect any adsorbents or contamination in the surface; and, as mentioned in the manuscript, before performing all these PPC experiments, our samples were kept at 400 K for at least one day in high vacuum ($\sim 10^{-6}$ torr) in order to remove all possible adsorbents (a typically way to clean off surface of vdW materials). Moreover, our MoS₂ sample shows the PPC effect with a time constant of $\sim 10^5 \text{ s}$ (Fig. 4), more than three order of magnitude longer than that in the α -In₂Se₃ nanosheets ($\sim 4 \text{ s}$, Fig. R3a, which is barely a PPC)⁸, indicating distinct mechanisms behind the PPC effect.

Finally, such discussion in the pre-PPC drop does not affect the analysis of the PPC and the extraction of the PPC time constant, because the pre-PPC drop and the PPC kinetics can be, and indeed were, deconvoluted from each other in our work by the baseline subtraction.

Figure R3. (a) and (b) Transient photocurrent curve of an α -In₂Se₃ nanosheet and its resultant schematic of surface depletion model (*Nanotechnology, 2020, 31.46, 465201*).⁸

Q3: *There is also some other evidence showing that this abrupt drop cannot be the minority recombination process. For example, in the reference 13 (Advanced materials 2013, 25 (25), 3456-3461.) that the authors provides in the last response letter, Fig.4d shows that the photocurrent is independent of temperature when $T < 200\text{K}$. The normal SRH model cannot explain this phenomenon because the SRH process is highly dependent on temperature (<https://www.nrel.gov/docs/fy99osti/25482.pdf>).*

Response:

The *Advanced materials 2013, 25 (25), 3456*, cited in our last response letter, indeed observed an abrupt drop of photocurrent in CVD grown MoS_2^9 , but did not explain or explore its origin, hence not excluding the possibility of minority recombination (although it is not our responsibility to explain the data of other people's).

First, what this work studied was photocurrent properties of CVD grown MoS_2^9 . It is well known that defect condition of CVD grown TMDs could be significantly different from that in mechanically exfoliated ones, as studied for numerous times by STEM in the literature¹⁰. CVD-grown monolayer TMDs often adsorb catalysts or contamination during the growth process, showing different optoelectronic properties from exfoliated ones.

The Shockley–Read–Hall (SRH) model describes the trap-assisted non-radiative recombination, that is, electrons transitioning between conduction (E_c) and valence (E_v) bands mediated by localized deep states (E_t)¹¹. In stark contrast to the DX center model (Fig. 3c), however, such transition of electrons between bands and deep traps does not need to overcome an energy barrier, hence leading to much shorter carrier lifetime and absence of the PERSISTENT photoconductivity (PPC) that we observed^{12,13}.

Q4: *Lastly, the reviewer acknowledges that it is unfair to ask the authors to adopt one model and reject others in this manuscript. But the authors should seriously discuss and acknowledge the possibility.*

Response:

As we responded in previously paragraphs, we respectfully disagree with the reviewer on the validity of the surface depletion model to explain our data, and believe our explanation is the most probable one taken into account all data and evidences. However, we agree with the reviewer that his/her comment is worth of mentioning in analyzing our work. We therefore modified our manuscript to reflect this discussion.

We have added the discussion on the possibility of the surface depletion model in note 7 of the Supplementary Information.

- 1 He, J. *et al.* Explicit Gain Equations for Single Crystalline Photoconductors. *ACS nano* **14**, 3405-3413 (2020).
- 2 Sett, S., Ghatak, A., Sharma, D., Kumar, G. P. & Raychaudhuri, A. Broad band single germanium nanowire photodetectors with surface oxide-controlled high optical gain. *The Journal of Physical Chemistry C* **122**, 8564-8572 (2018).
- 3 Zhao, X., Tu, P., He, J., Zhu, H. & Dan, Y. Cryogenically probing the surface trap states of single nanowires passivated with self-assembled molecular monolayers. *Nanoscale*

- 10**, 82-86 (2018).
- 4 Tachikawa, M. *et al.* Observation of the persistent photoconductivity due to the DX center in GaAs under hydrostatic pressure. *Japanese journal of applied physics* **24**, L893 (1985).
- 5 Li, J. *et al.* Nature of Mg impurities in GaN. *Applied physics letters* **69**, 1474-1476 (1996).
- 6 Amani, M. *et al.* Near-unity photoluminescence quantum yield in MoS₂. *Science* **350**, 1065-1068 (2015).
- 7 Wang, H., Zhang, C. & Rana, F. Surface recombination limited lifetimes of photoexcited carriers in few-layer transition metal dichalcogenide MoS₂. *Nano letters* **15**, 8204-8210 (2015).
- 8 Anandan, M. *et al.* High-responsivity broad-band sensing and photoconduction mechanism in direct-gap α -In₂Se₃ nanosheet photodetectors. *Nanotechnology* **31**, 465201 (2020).
- 9 Zhang, W. *et al.* High - gain phototransistors based on a CVD MoS₂ monolayer. *Advanced materials* **25**, 3456-3461 (2013).
- 10 Hong, J. *et al.* Exploring atomic defects in molybdenum disulphide monolayers. *Nature communications* **6**, 1-8 (2015).
- 11 Shockley, W. & Read Jr, W. Statistics of the recombinations of holes and electrons. *Physical review* **87**, 835 (1952).
- 12 Fossum, J., Mertens, R., Lee, D. & Nijs, J. Carrier recombination and lifetime in highly doped silicon. *Solid-state electronics* **26**, 569-576 (1983).
- 13 McCluskey, M. D. & Haller, E. E. *Dopants and defects in semiconductors*. (CRC press, 2018).

REVIEWERS' COMMENTS

Reviewer #3 (Remarks to the Author):

The reviewer has no more questions and recommends to publish the manuscript as it is.

REVIEWERS' COMMENTS

Reviewer #3 (Remarks to the Author):

The reviewer has no more questions and recommends to publish the manuscript as it is.

Response:

We thank the reviewer #3 for recommendation this manuscript for publication.